# Molecular mechanisms of receptor recognition and antibody neutralization of coxsackievirus A6

Xianliang Ke[1,3], Xue Li[2,3], Zeyu Liu[2,3], Kexin Liu[2], Weichi Liu[1], Xingyu Yan[2], Bo Shu[1] ✉ & Chao Zhang [2] ✉

Coxsackievirus A6 (CVA6), a major cause of hand, foot, and mouth disease, lacks approved vaccines or drugs. KRM1 is its only known receptor, but its precise role remains unclear. This study investigates CVA6's entry mechanism and antibody neutralization. Cryo-EM shows CVA6 clinical strain HeB primarily exists as mature virions. KRM1 binding within the canyon triggers conversion to uncoating intermediate, defining KRM1 as an uncoating receptor for CVA6. However, *KRM1* knockout reduces CVA6 infectivity without affecting attachment. Conversely, disrupting heparan sulfate proteoglycan (HSPG) impairs both viral attachment and infectivity, and CVA6 virions bind heparin directly. These results support a two-receptor entry model for CVA6: HSPG mediates viral attachment, while KRM1 induces uncoating. Additionally, we develop two CVA6-specific protective antibodies (1F4 and 3H7), targeting a new antigenic site near the three-fold axis of the viral capsid. These antibodies sterically block KRM1 binding and function post-attachment, consistent with KRM1's role. The findings elucidate CVA6 entry and offer a basis for antibody interventions.

Hand, foot, and mouth disease (HFMD), a highly contagious viral illness, has become a significant global public health challenge, due to frequent large-scale epidemics across Asia and other regions[1]. HFMD is mainly caused by species A enteroviruses (EV-A) within the *Picornaviridae* family. Historical HFMD outbreaks have been predominantly associated with enterovirus A71 (EV-A71) and coxsackievirus A16 (CVA16). However, recent epidemiological surveys indicate that coxsackievirus A6 (CVA6) has rapidly emerged as the leading causative agent of HFMD in many countries, surpassing EV-A71 and CVA16[2–6]. Unlike classic HFMD, CVA6 infections are frequently associated with atypical clinical features, such as widespread vesiculobullous eruptions, onychomadesis, and severe complications like meningitis, myocardial injury, and pulmonary edema[1,7]. This highlights the urgent need for research on CVA6's pathogenic mechanisms and effective countermeasures.

CVA6 is a nonenveloped RNA virus with a ~ 30 nm icosahedral capsid composed of 60 protomers. Cryo-electron microscopy (cryo-EM) studies reveal that CVA6 has three distinct particle forms: mature virions, uncoating intermediate A (altered) particles, and empty particles[8,9]. CVA6 virions contain the infectious RNA genome within a compact capsid composed of VP1, VP2, VP3, and VP4. A-particles have viral RNA and an expanded capsid lacking VP4. Empty particles have an expanded capsid of VP1, VP3, and uncleaved VP0, but no viral RNA[8,9]. All three particle forms share key structural features: a star-shaped protrusion at the five-fold axis, surrounded by a deep depression (canyon), and a protrusion at the three-fold axis. In mature virions, VP1 hydrophobic pocket beneath the canyon contains a lipid molecule termed pocket factor, which stabilizes the capsid[8]. This pocket is empty in A-particles and empty particles[8,9]. Unlike most enteroviruses where mature virions are predominant[10,11], studies of purified CVA6

[1]State Key Laboratory of Virology and Biosafety, Wuhan Institute of Virology, Center for Biosafety Mega-Science, Chinese Academy of Sciences, Wuhan, Hubei, China. [2]Shanghai Institute of Infectious Disease and Biosecurity, Fudan University, Shanghai, China. [3]These authors contributed equally: Xianliang Ke, Xue Li, Zeyu Liu. ✉e-mail: shubo@wh.iov.cn; chao_zhang@fudan.edu.cn

particles (prototype strain Gdula and clinical strain TW-141) reveal that A-particles are the major component, with virions being rare (1.2% for Gdula; 0% for TW-141)[8,9]. This led to the view that A-particles represent CVA6's primary infectious form and are the optimal vaccine target[9]. However, this conclusion is based on a limited number of examples and may not apply to all CVA6 strains. Further research is necessary to determine whether mature virions or A-particles dominate in other CVA6 strains.

Many enteroviruses utilize a two-receptor entry mechanism: an attachment receptor mediates initial cell binding, followed by an uncoating receptor triggering conformational changes and genome release[12–14]. Only one cellular receptor has been identified for CVA6: Kringle-containing transmembrane protein 1 (KREMEN1 or KRM1). KRM1 also serves as the functional receptor for CVA2, CVA3, CVA4, CVA5, CVA8, CVA10, and CVA12[15,16]. Previous studies focused on the interaction between KRM1 and CVA10 and reveal that soluble KRM1 binds CVA10 virions with high affinity and induces their transition to uncoating intermediate A-particles[15,17,18]. However, KRM1's precise role in CVA6 infection remains undefined. Moreover, whether CVA6 utilizes alternative receptors is unknown.

Currently, no approved vaccines or drugs are available for CVA6-related HFMD. Monoclonal antibodies (MAb) represent promising candidates for antiviral development. Yet only one CVA6-specific neutralizing MAb (termed 1D5) has been reported, targeting the five-fold vertex to block viral attachment[9]. However, other antigenic sites on the CVA6 capsid and distinct neutralization mechanisms likely exist, underscoring the need to develop additional CVA6-neutralizing MAbs.

In this study, we systematically elucidate the molecular mechanisms of CVA6 receptor recognition and antibody neutralization. We show that purified viral particles of CVA6 clinical strain HeB (strain 54203/HeB/2012, isolated from an HFMD patient in Hebei Province, China, 2012) predominantly exist as mature virions. Crucially, KRM1 binding within the viral canyon region induces structural conversion to A-particles, establishing KRM1 as the uncoating receptor for CVA6. *KRM1* knockout inhibits CVA6 infection without affecting viral attachment, whereas HSPG knockout reduces both viral infection and attachment. These results indicate that HSPG is the primary attachment receptor for CVA6, while KRM1 is essential for post-attachment uncoating. Furthermore, we developed two potent CVA6-specific neutralizing and protective MAbs, 1F4 and 3H7. Both MAbs target the new antigenic site located near the three-fold axis of viral capsid. They sterically block KRM1-virion interactions and act primarily at post-attachment stages, consistent with KRM1's uncoating role. Collectively, these findings establish a two-receptor entry mechanism for CVA6 and reveal the structural basis for therapeutic antibody intervention.

## Results

### Development and characterization of neutralizing MAbs against CVA6

To generate CVA6-specific neutralizing MAbs, hybridomas were produced from mice immunized with purified CVA6-TW-141 (CVA6-141) virions. These hybridomas were subsequently screened for neutralizing activity against the CVA6-141 strain. Ultimately, two hybridoma clones (1F4 and 3H7; IgG2a isotype) demonstrating potent neutralizing activity were isolated (Fig. 1a). MAbs 1F4 and 3H7 were purified, and their neutralizing activity against both CVA6-141 and CVA6-HeB strains was quantified using two methods: (1) the minimal neutralizing concentration (MNC), defined as the lowest antibody concentration that completely prevented cytopathic effect (CPE) in microscopic observation assays, and (2) the half-maximal inhibitory concentration (IC50), calculated from dose-response curves in cell viability assays. As shown in Fig. 1a–c, MAb 1F4 exhibited potent neutralization against both the CVA6-141 strain (MNC: 0.313 μg/mL; IC50: 0.052 μg/mL) and

the CVA6-HeB strain (MNC: 0.625 μg/mL; IC50: 0.193 μg/mL). Similarly, MAb 3H7 demonstrated strong neutralizing activity against CVA6-141 (MNC: 0.625 μg/mL; IC50: 0.103 μg/mL) and CVA6-HeB (MNC: 1.25 μg/mL; IC50: 0.387 μg/mL). In contrast, the IgG control antibody 3A2[19] showed no neutralization at any tested concentration. Together, these results indicate that both MAbs (1F4 and 3H7) display comparable neutralization potency against CVA6 strains. Furthermore, neither MAb neutralized CVA10 strain S0148b even at the highest concentration tested (10 μg/mL) (Fig. 1a), confirming their specificity for CVA6.

The MAbs were tested by ELISA for their ability to recognize purified CVA6-HeB virions. The HeB strain of CVA6 was selected for subsequent experiments due to its higher proportion of mature virions compared to the 141 strain, which predominantly produces A-particles as documented[9]. Both anti-CVA6 MAbs 1F4 and 3H7 specifically bound to CVA6-HeB virions, whereas the IgG control antibody showed no reactivity (Fig. 1d). Notably, MAb 3H7 exhibited significantly stronger binding to CVA6-HeB virions compared to MAb 1F4. To quantify this difference, we measured the binding affinity of the MAbs to CVA6-HeB virions using bio-layer interferometry (BLI). Consistent with the ELISA results, BLI revealed that MAb 3H7 displayed a higher binding affinity (KD = 0.40 nM) than MAb 1F4 (KD = 3.43 nM) (Fig. 1e).

The protective efficacy of MAbs 1F4 and 3H7 was evaluated in the established neonatal mouse models of CVA6 infection, using two distinct highly lethal strains: CVA6-HeB[20] and CVA6-S0087b[21,22]. Note that CVA6-S0087b exhibited high virulence in neonatal mice, despite failing to produce detectable CPE in vitro. For the prophylactic evaluation, one-day-old ICR mice (n ≥ 11 per group) were randomly assigned to receive PBS, 10 mg/kg of anti-CVA6 MAbs (1F4 or 3H7), or 10 mg/kg of control IgG 3A2[19]. Twenty-four hours after treatment, the mice were challenged with either CVA6-HeB or CVA6-S0087b. Survival rates were monitored daily and are summarized in Supplementary Fig. 1. Following CVA6-HeB challenge, PBS- and control IgG-treated groups developed progressive limb weakness and paralysis, culminating in 100% mortality within 10 dpi. Strikingly, 10 of 11 mice administered 1F4 and all 3H7-treated mice survived the infection. A parallel trend was observed in the CVA6-S0087b challenge model: all PBS- and control IgG-treated mice succumbed to infection by 7 dpi, whereas both 1F4- and 3H7-treated mice remained fully protected. These results conclusively demonstrate the robust in vivo prophylactic activity of 1F4 and 3H7 against divergent CVA6 strains.

To evaluate therapeutic efficacy, two-day-old ICR mice (n ≥ 11 per group) were infected with CVA6-HeB or CVA6-S0087b and 24 hours later treated with PBS or anti-CVA6 MAbs (1F4 or 3H7; 10 mg/kg). Survival rates were monitored daily (Fig. 1f). Following CVA6-HeB infection, all PBS-treated mice succumbed to infection within 8 dpi, while 10/13 mice in the 1F4 group and all 3H7-treated mice survived until 14 dpi. A parallel protective trend was observed against CVA6-S0087b: both 1F4 and 3H7 MAbs conferred 100% survival in treated mice, whereas 100% mortality was observed in the PBS-treated control group. These findings highlight the significant therapeutic efficacy of 1F4 and 3H7 MAbs against distinct CVA6 strains. Taken together, these results demonstrate robust in vivo efficacy of these MAbs in both pre- and post-exposure scenarios.

### Neutralization mechanisms of MAbs 1F4 and 3H7

To investigate the neutralization mechanisms of MAbs 1F4 and 3H7 against CVA6, time-of-addition assays were performed to identify the specific viral lifecycle stage targeted by these antibodies. In this assay, MAbs were assessed under: (1) Pre-attachment: antibody-virus (CVA6-HeB) premixes were adsorbed onto cells at 4 °C prior to 37 °C infection; (2) Post-attachment: antibodies were administered at 0 h or 0.5 h after shifting virus-bound cells to 37 °C. Viral RNA levels were quantified at 6 hpi by real-time quantitative reverse transcription PCR (RT-qPCR). As shown in Fig. 2a, MAb 1F4 exhibited potent

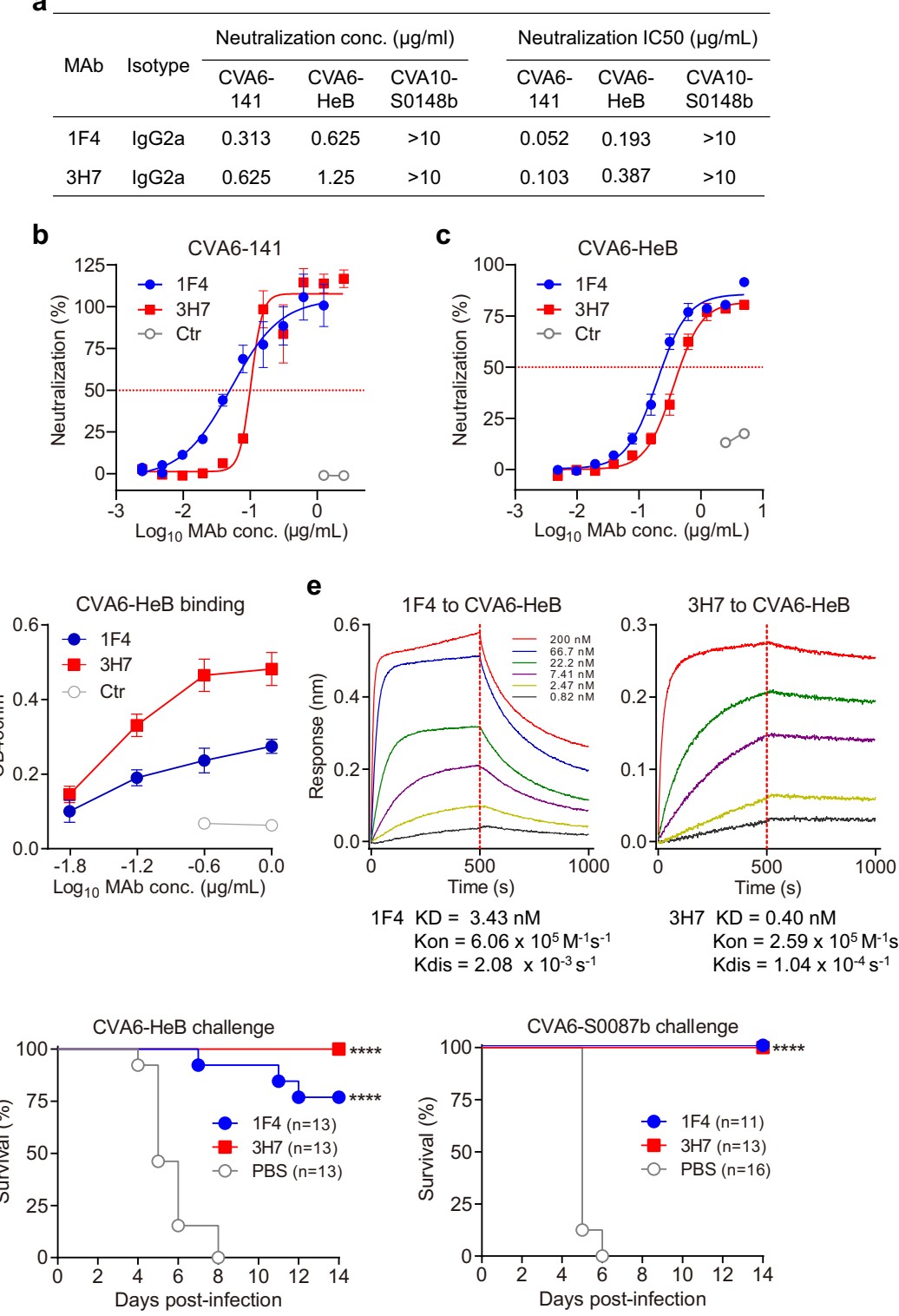

**a**

| MAb | Isotype | Neutralization conc. (µg/ml) | | | Neutralization IC50 (µg/mL) | | |
|---|---|---|---|---|---|---|---|
| | | CVA6-141 | CVA6-HeB | CVA10-S0148b | CVA6-141 | CVA6-HeB | CVA10-S0148b |
| 1F4 | IgG2a | 0.313 | 0.625 | >10 | 0.052 | 0.193 | >10 |
| 3H7 | IgG2a | 0.625 | 1.25 | >10 | 0.103 | 0.387 | >10 |

**e**

1F4 KD = 3.43 nM
Kon = 6.06 x 10^5 M^-1 s^-1
Kdis = 2.08 x 10^-3 s^-1

3H7 KD = 0.40 nM
Kon = 2.59 x 10^5 M^-1 s^-1
Kdis = 1.04 x 10^-4 s^-1

inhibition in both pre-attachment (Pre, 33% residual viral RNA) and immediate post-temperature shift (Post-0h, 33% residual RNA) regimens, but efficacy sharply diminished at Post-0.5 h (82% residual RNA), defining a narrow temporal window for blocking early post-attachment entry. MAb 3H7 showed robust inhibition at pre-attachment (1% residual viral RNA) and Post-0h (17% residual RNA), with reduced activity at Post-0.5 h (87% residual RNA),

demonstrating neutralization capacity spanning both viral attachment and early post-attachment entry steps.

To determine whether MAbs 1F4 or 3H7 interfere with the initial attachment phase of CVA6 entry, we pre-incubated CVA6-HeB virions with antibodies prior to adsorption onto pre-chilled RD cells at 4 °C. After washing, cell-bound viral RNA was measured by RT-qPCR. 1F4 and control IgG showed no inhibition at any tested dose, while the 3H7

**Fig. 1 | Development and characterization of anti-CVA6 neutralizing MAbs.**
**a** Isotype and neutralization activity of MAbs 1F4 and 3H7. Neutralization concentration was defined as the lowest antibody concentration fully preventing cytopathic effect. Neutralization $IC_{50}$ values were derived from cell viability assays (**b**, **c**). **b**, **c** Neutralization potency of the MAbs against CVA6 strains 141 and HeB. Dose-response curves of the MAbs against CVA6-141 (**b**) and CVA6-HeB (**c**) were quantified via cell viability assays. An anti-SARS-CoV-2 antibody served as negative control (Ctr). Data represent mean ± SEM of four replicate wells in 96-well cell culture plates. Conc., concentration. **d** Binding of the MAbs to purified CVA6-HeB virions analyzed by ELISA. Data are expressed as mean ± SD from triplicate wells. **e** Binding kinetic analysis of the MAbs to immobilized CVA6-HeB virions by biolayer

interferometry (BLI). Association and dissociation steps are divided by dotted red line. Sensorgrams show responses at serial MAb concentrations (nM range labeled). Equilibrium dissociation constants (KD) were calculated by Octet Data Analysis software. **f** Therapeutic efficacy of MAbs 1F4 and 3H7 against CVA6-HeB and CVA6-S0087b lethal infections in mice. Suckling mice received PBS, MAb 1F4 or 3H7 (10 μg/g) 24 hours post-infection with CVA6-HeB or CVA6-S0087b. Survival curves comparing antibody-treated groups with PBS controls are shown. Statistical significance was determined by the Log-rank (Mantel−Cox) test. ****, $p < 0.0001$. The number of mice in each group is shown in parentheses. Source data are provided as a Source Data file.

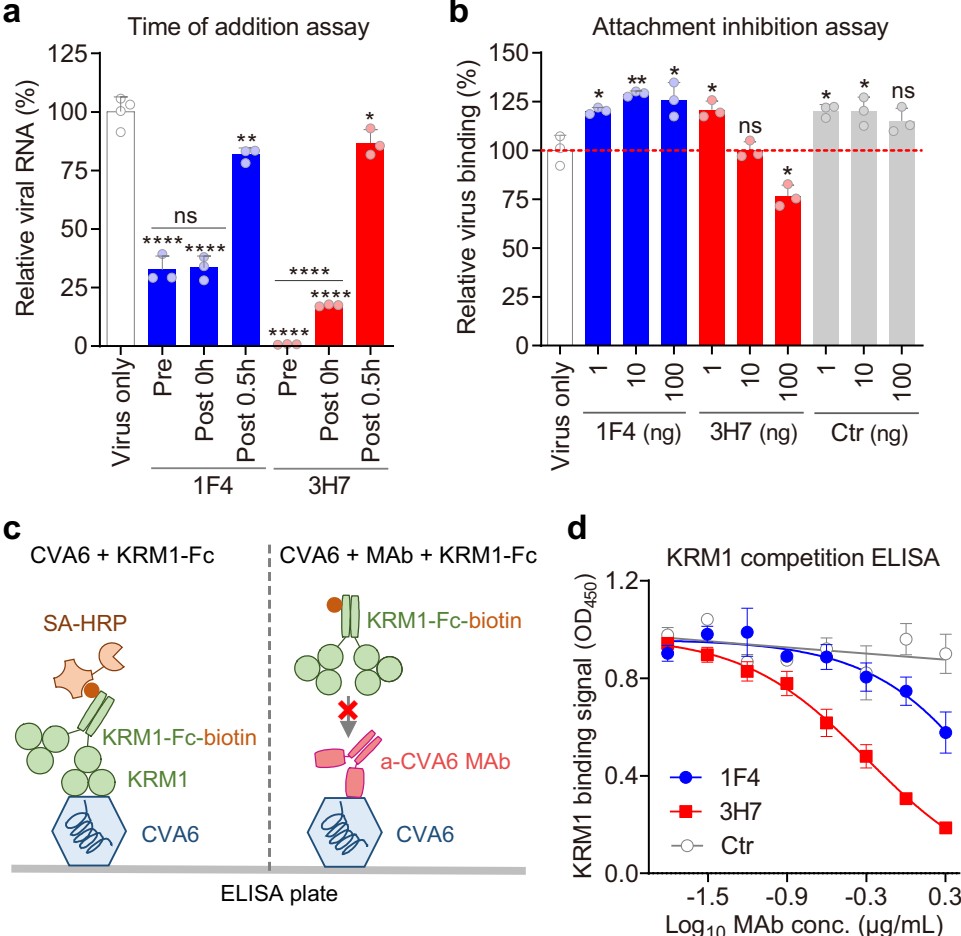

**Fig. 2 | Mechanisms of action of CVA6-neutralizing MAbs. a** Time of addition assay. CVA6-HeB was incubated with MAbs 1F4 or 3H7 before (Pre) or at indicated time points after (Post) viral attachment to RD cells. Viral RNA was quantified by RT-qPCR at 6 hours post-infection (hpi) and is expressed as a percentage of the virus-only control. Data are mean ± SD (n = 4 for virus-only; n = 3 for others). $p = 0.0055$ (1F4-Post-0.5 h); $p = 0.0336$ (3H7-Post-0.5 h). **b** Attachment inhibition assay. CVA6-HeB was incubated with serially diluted anti-CVA6 MAbs (1F4, 3H7) or a control antibody for 1 h, followed by adsorption to pre-chilled RD cells at 4 °C for 2 h. Cell-associated viral RNA was analyzed via RT-qPCR. Viral RNA levels are expressed as percentages relative to the virus-only group. and is expressed as a percentage of the virus-only control. Data are mean ± SD (n = 3). $p = 0.0112$ (1F4-1ng), $p = 0.0030$ (1F4-10ng), $p = 0.0181$ (1F4-100ng), $p = 0.0152$ (3H7-1ng), $p = 0.0126$ (3H7-100ng).

Statistical note for **a**, **b**: Significance was determined by two-tailed Student's t-test versus the virus-only control: ns, $p \geq 0.05$; *, $p < 0.05$; **, $p < 0.01$; ****, $p < 0.0001$. **c** Schematic representation of competitive binding ELISA. (Left) KRM1 binding: Biotinylated KRM1-Fc protein binds to immobilized CVA6 virions, detected by HRP-conjugated streptavidin (SA-HRP). (Right) Antibody blockade: Anti-CVA6 MAbs bind to virions, sterically hindering KRM1-Fc binding and reducing detection signal. **d** KRM1 competition ELISA. Serially diluted anti-CVA6 MAbs (1F4, 3H7) or control antibody were tested for blocking biotinylated KRM1-Fc binding to immobilized CVA6-HeB virions. Bound KRM1-Fc was detected using HRP-streptavidin. Data represent mean ± SEM of triplicate wells. Source data are provided as a Source Data file.

antibody demonstrated weak suppression of viral attachment only at the highest concentration (100 ng/well) (Fig. 2b). These observations suggest that the neutralizing mechanisms of both 1F4 and 3H7 predominantly target post-attachment steps.

Human KRM1 mediates CVA6 infection[15]. To investigate whether MAbs 1F4 and 3H7 interfere with CVA6-KRM1 interactions, we

developed a competitive receptor-binding ELISA (Fig. 2c). Immobilized CVA6-HeB virions were incubated with biotinylated KRM1-Fc fusion protein in the presence of IgG control, 1F4, or 3H7. Bound KRM1-Fc was detected using HRP-streptavidin (SA-HRP). Compared with IgG control, both 1F4 and 3H7 significantly inhibited receptor-virus interaction, with 3H7 showing stronger blockade (Fig. 2d). These data suggest

that 1F4 and 3H7 may sterically hinder CVA6 engagement with membrane-associated KRM1.

Both 1F4 and 3H7 MAbs effectively blocked CVA6-KRM1 binding but had little effect on viral attachment to cells. Notably, both antibodies potently neutralized the virus post-attachment (Fig. 2). These results support the hypothesis that KRM1 is not essential for initial CVA6 attachment but is critical for post-attachment entry steps.

## KRM1 binding triggers uncoating of mature CVA6 virions

Previous studies identified KRM1 as essential for CVA6 infection[15]. To validate the infection dependency, we infected wild-type RD cells and their *KRM1*-knockout (Δ*KRM1*) counterparts (generated via CRISPR-Cas9[16]) with CVA6-HeB. Quantification of viral titers at 24 hours post-infection (hpi) revealed a 36-fold reduction in Δ*KRM1* cells compared to wild-type RD cells (Supplementary Fig. 2a), confirming KRM1's essential role in productive CVA6 infection. Importantly, ELISA-based binding assay showed that immobilized CVA6-HeB virions exhibited dose-dependent binding to human KRM1-Fc, whereas ACE2-Fc controls showed no binding activity (Supplementary Fig. 2b), providing the first experimental evidence of direct physical interaction between CVA6 and KRM1. These findings established KRM1 as both a physical binding partner and functional receptor, prompting us to investigate its structural impact.

To determine whether KRM1 binding induces structural reorganization in CVA6, we compared cryo-EM structures of CVA6-HeB virus particles before and after KRM1 receptor engagement (Supplementary Tables 1 and 2). For virus-only analysis, UV-inactivated CVA6-HeB viral particles were analyzed by cryo-EM, revealing two populations: (1) mature virions (93.0%, 20,204 particles); (2) empty particles (7.0%, 1531 particles) (Supplementary Fig. 3). The CVA6-HeB mature virion structure (2.52 Å resolution) exhibits a compact capsid (~310 Å diameter) with closed two-fold axis channels, pocket factor (modeled as stearic acid) in VP1 hydrophobic pocket, and retained genomic RNA (Fig. 3a, c, e, g). The CVA6-HeB empty particle structure (3.47 Å resolution) shows expanded architecture (~317 Å diameter), open two-fold channels, loss of pocket factor, and no viral RNA (Fig. 3b, d, f, h). In both maps, densities corresponding to the residue backbones and side chains—particularly the bulky ones—were well resolved and readily identifiable (Supplementary Fig. 4). Structural alignment with CVA6 prototype strain Gdula[8] confirmed structural conservation, with overall root-mean-square deviation (RMSD) values of 0.3 Å for mature virions and 0.5 Å for empty particles (Supplementary Fig. 5). Notably, no A-particles were detected in CVA6-HeB samples (Supplementary Fig. 3), contrasting earlier reports of A-particle-dominated CVA6-Gdula and CVA6-141 preparations[8,9].

For receptor interaction studies, purified CVA6-HeB virus particles were incubated with recombinant His-tagged human KRM1 ectodomain at a 1:120 molar ratio (35 min at room temperature) and subjected to cryo-EM analysis. Three distinct populations were resolved: (1) KRM1-bound mature virions (28.7%, 24,506 particles); (2) A-particles (69.8%, 59,654 particles); (3) empty particles (1.5%, 1290 particles) (Supplementary Fig. 6). The KRM1-virion complex structure (2.55 Å resolution) retains mature virion architecture with closed two-fold axis channels, VP1 pocket factor and intact RNA, while displaying clear KRM1 density on the capsid surface (Fig. 3i, l, o, r). The A-particle structure (2.49 Å resolution) displays expanded capsids with open two-fold channels, an ejected pocket factor, and retained RNA, with only residual KRM1 density (Fig. 3j, m, p, s). The clear and well-fitted densities of both the residue backbones and side chains enabled the determination of the structures of viral particles at atomic or near-atomic resolution (Supplementary Fig. 7). Structural alignment with A-particles of CVA6 prototype strain Gdula[8] and clinical strain 141[9] confirmed high similarity, with overall RMSD values of 0.5 and 1.0 Å, respectively (Supplementary Fig. 8). The empty particle structure

(3.17 Å resolution) remains structurally identical to the untreated empty particle (Fig. 3k, n, q, t).

Compared to virus-only preparations, KRM1 treatment induced a striking redistribution of particle populations: mature virion proportions decreased from >90% to <30%, while A-particles − absent in untreated virus samples − emerged as the dominant species (70%). Empty particle levels remained consistently low (<10%) throughout the process (Fig. 3). This receptor-dependent shift demonstrates that A-particles originate from structural remodeling of mature virions upon KRM1 binding. Crucially, KRM1 drives this conversion with high efficiency under neutral pH at room temperature (35-min incubation), establishing it as the physiological uncoating receptor for CVA6.

## Structural analysis of CVA6-KRM1 complex

The cryo-EM structure of KRM1-bound CVA6 virion (Fig. 3i) reveals the interaction interface between CVA6 and KRM1. KRM1 binds near each five-fold vertex within the canyon, spanning the VP1, VP2, and VP3 proteins of two adjacent protomers (Fig. 4a, b). Each KRM1 molecule engages both protomers, burying 741.7 Å$^2$ on protomer 1 and 775.6 Å$^2$ on protomer 2, resulting in a total interface area of 1517.3 Å$^2$ (Fig. 4b).

The KRM1 ectodomain comprises three structural domains: the N-terminal Kringle (KR) domain, the WSC domain, and the CUB domain. The interaction with CVA6 is mediated through the KR and WSC domains of KRM1 (Fig. 4c, d). CVA6 protomer 1 forms extensive contacts with the WSC domain (Supplementary Table 3). Specifically, the R90 residue in VP3 CD loop forms a hydrogen bond with WSC residue T138 (Fig. 4e); the VP3 C-terminal residues D231 and Q234 form hydrogen bonds with WSC residues N128 and Y178, respectively (Fig. 4f); A dense hydrogen-bond network links VP1 C-terminal residues D284, A286, D292, and E294 to the WSC region T135−T141 (Fig. 4g). CVA6 protomer 2 interacts with both the KR and WSC domains of KRM1 (Supplementary Table 3). Specifically, the K157 and D159 residues in VP1 EF-loop form salt bridges and hydrogen bonds with WSC residues D201 and H126, respectively (Fig. 4h); the Q211 residue in VP1 GH-loop forms a hydrogen bond with WSC residue G192 (Fig. 4i); the K140 and N142 residues in VP2 EF-loop establish hydrogen bonds with KR residues G89 and D90; VP2 K140 additionally forms salt bridges with KR residues D88 and D90; VP2 K140 also forms π-cation interactions with the aromatic KR residues W94 and W106 (Fig. 4j). Notably, VP2 K140 is fully conserved in KRM1-dependent enteroviruses (including CVA6) and is critical for receptor binding and infectivity[16]. The interaction of K140 with KRM1 residues D88, G89, D90, W94, and W106 is conserved not only in CVA6 and CVA10 (Supplementary Fig. 9)[17], but also potentially in other KRM1-dependent enteroviruses.

The KRM1-bound CVA6 retains a mature virion architecture nearly identical to its unbound (apo) state, with a global RMSD value of 0.4 Å (Supplementary Fig. 10a). Detailed structural comparisons reveal only minor deviations in specific regions: VP2 EF-loop (RMSD = 0.4 Å), VP3 C-terminus (0.4 Å), and VP3 GH-loop (0.5 Å) (Supplementary Fig. 10b). Prior studies on KRM1-bound CVA10 [17] reported a similarly mature virion structure but with larger conformational changes in the same regions: VP2 EF-loop (RMSD = 1.4 Å), VP3 C-terminus (0.7 Å), and VP3 GH-loop (2.3 Å) (Supplementary Fig. 10c, d). These differences in structural variability may stem from inherent variations between the viral serotypes or from differences in the experimental methods used.

## Cryo-EM structure of CVA6 in complex with 1F4 Fab reveals neutralization mechanism

To elucidate the neutralizing mechanism of the 1F4 antibody against CVA6, we determined the cryo-EM structure of CVA6 complexed with 1F4 Fab (Supplementary Table 4). Incubation of purified CVA6-HeB viral particles with excess 1F4 Fab yielded two distinct populations: (1) 1F4 Fab-bound mature virions (98.9%); (2) empty particles (1.1%) (Supplementary Fig. 11). The well-fitted backbones and side chains (Supplementary Fig. 13a−c) allowed us to

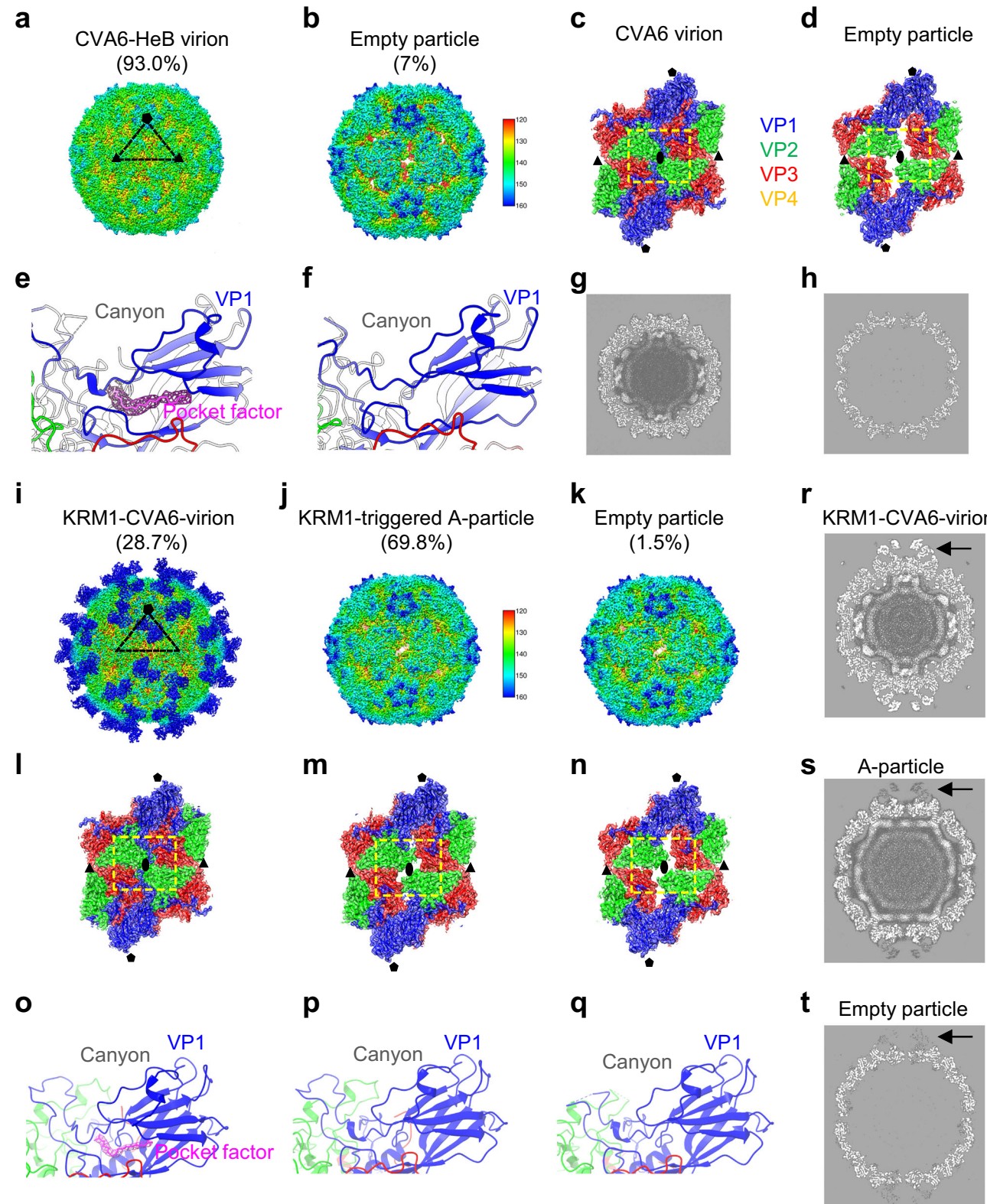

determine the high-resolution structure of the 1F4–virion complex at 2.11 Å, which preserves characteristic features of mature virions, including closed two-fold axis channels, pocket factor in VP1, and intact genomic RNA. Notably, clear Fab density is observed on the capsid surface (Fig. 5a, c, e, g). Structural comparison with unbound CVA6 virions reveals remarkable similarity, with an overall RMSD of 0.4 Å. The empty particle structure, resolved at 3.12 Å resolution, is identical to untreated empty particle (Fig. 3b),

exhibiting capsid expansion without detectable pocket factor, RNA or Fab density (Fig. 5b, d, f, h). Notably, 1F4 Fab treatment does not significantly alter the population distribution between mature virions (>90%) and empty particles (<10%) compared to virus-only pre-parations (Fig. 3a, b). These findings demonstrate that 1F4 selectively binds to mature virions rather than empty particles, and that 1F4 Fab binding does not induce significant conformational changes in the viral capsid.

**Fig. 3 | Cryo-EM structures of CVA6 particles before and after KRM1 binding.** **a**, **b** Cryo-EM density maps of UV-inactivated CVA6-HeB mature virion (**a**) and empty particle (**b**). All maps are radially colored (scale in Å) and viewed along the two-fold axis. One icosahedral asymmetric unit is marked by a black triangle. Pentagons and triangles denote five-fold and three-fold symmetry axes, respectively. 2D classification revealed 93.0% mature virions and 7.0% empty particles. **c**, **d** Twofold axis channel structures in the CVA6 mature virion (**c**) and empty particle (**d**). VP1 (blue), VP2 (green), VP3 (red), and VP4 (yellow) are shown; this color scheme applies to all panels. Ellipses represent the two-fold axis. Conformational differences are outlined by yellow dashed rectangles. **e**, **f** Atomic models of the VP1 hydrophobic pocket in the CVA6 mature virion (**e**) and empty particle (**f**). A pocket factor (magenta stick) is fitted into the VP1 hydrophobic pocket of the CVA6 virion, with corresponding cryo-EM density (magenta mesh). **g**, **h** Central cross-sectional views (Z-axis) of CVA6 mature virion (**g**) and empty particle (**h**). **i**–**k** Cryo-EM analysis of the KRM1-treated CVA6 sample revealed a shifted composition: 28.7% KRM1-bound virions (**i**), 69.8% A-particles (**j**), and 1.5% empty particles (**k**). **l**–**n** Twofold axis channel structures in the KRM1-bound CVA6 virion (**l**), A-particle (**m**), and empty particle (**n**). **o**–**q** VP1 pocket models for KRM1-bound CVA6 virion (**o**), A-particle (**p**), and empty particle (**q**). The pocket factor (magenta) is present in KRM1-bound virion but absent in A-particle and empty particle. KRM1 density is omitted for clarity. **r**–**t** Central cross-sectional views of the KRM1-bound CVA6 virion (**r**), A-particle (**s**), and empty particle (**t**) density maps. Black arrows indicate KRM1 receptor density. Central sections are displayed without masking or sharpening to preserve the native density information.

The cryo-EM structure of the 1F4-virion complex shows symmetric antibody binding, with three 1F4 Fab molecules arranged around each three-fold axis of viral capsid (Fig. 5a). Each 1F4 Fab binds to a single protomer by interacting with the VP3 AB-loop (also termed "knob", residues 56–65) and the VP1 C-terminus (Fig. 5i-j, Supplementary Table 5). The VP3 knob inserts into a cleft formed by the heavy chain complementarity-determining region 1 (CDR1) and 3 (CDR3) loops, as well as the light chain CDR1 of 1F4 (Fig. 5j). Key interactions include hydrogen bonds between: (1) VP3 knob residues G60, T62, and S65 and heavy chain CDR1 residues T30, S31, and Y33; (2) VP3 residues N56 and T58 with heavy chain CDR3 residue N101; (3) VP3 residue T59 with light chain CDR1 residue N32 (Fig. 5j). The VP1 C-terminus further stabilizes the interface through hydrogen bonds: residue I288 interacts with heavy chain CDR3 residue N101, and S287/T289 engage light chain framework 3 (FR3) residue N53 (Fig. 5j). The interaction interface between CVA6 and 1F4 covers 842.4 Å$^2$ per protomer, with the 1F4 heavy and light chains contributing 61.8% and 38.2% of the buried areas, respectively (Supplementary Table 6).

To explore the spatial relationship between the 1F4 epitope and the KRM1 receptor-binding site, we conducted comparative footprint mapping on virion surfaces using RIVEM analysis (Fig. 5k). The analysis revealed that the VL (light chain variable region) domain of 1F4 directly overlaps with the KRM1 binding region, indicating a potential mechanism of receptor blockade mediated by the VL domain. Additionally, structural superposition of the KRM1-CVA6-virion complex and the 1F4-CVA6-virion complex demonstrated steric clashes between the 1F4 VL domain and the WSC and CUB domains of KRM1 (Fig. 5l). These structural analyses demonstrate that 1F4 can sterically hinder KRM1 from binding to the CVA6 virion, thereby elucidating the structural basis for its receptor-blocking activity (Fig. 2d).

### Cryo-EM structure of CVA6 in complex with 3H7 Fab
To determine the structural basis for CVA6 neutralization by the 3H7 antibody, we determined the cryo-EM structure of the CVA6–3H7 Fab complex (Supplementary Table 7). After incubation of purified CVA6 particles with excess 3H7 Fab, two major populations were observed: (1) 3H7 Fab-bound CVA6 virions (98.1%, resolved at 2.59 Å); (2) 3H7 Fab-decorated empty particles (1.9%, resolved at 3.26 Å) (Fig. 6a, b, Supplementary Fig. 12). Notably, 3H7 Fab binds both mature virions and empty particles, as evidenced by clear Fab densities on both particle types (Fig. 6a, b). The well-resolved backbones and side chains allowed us to determine the structures of both complexes (Supplementary Fig. 13d–i). The 3H7 Fab-bound CVA6 virions retain key features of mature virions, including a compact capsid with closed two-fold axis channels, the VP1 pocket factor, and well-resolved genomic RNA (Fig. 6a, c, e, g). When compared to unbound CVA6 virions, the overall structure shows remarkable similarity, with an RMSD of 0.3 Å. In contrast, the 3H7 Fab-decorated empty particles display an expanded architecture with open two-fold channels and lack both the VP1 pocket factor and internal RNA densities (Fig. 6b, d, f, h). In addition, the 3H7 Fab treatment did not change the population ratio of mature virions (>90%) to empty particles ( < 10%) compared to the untreated virus (Figs. 3a, b and 6a, b). These results indicate that 3H7 Fab binds to the virus without inducing significant conformational changes in the viral capsid.

Similar to 1F4, the 3H7 Fab binds close to the icosahedral three-fold axis on the viral capsid (Fig. 6a, b), with each Fab engaging a single protomer (Fig. 6i). Both heavy and light chains of 3H7 mediate interactions with VP1, VP2, and VP3 proteins within the same protomer (Fig. 6i). The 3H7 Fab buries 1124.4 Å$^2$ of surface area per protomer. Specifically, the 3H7 heavy and light chains account for 70.1% and 29.9% of the binding interactions, respectively (Supplementary Table 6). The VH (heavy chain variable region) domain establishes an extensive network of hydrogen bonds and salt bridges with the capsid (Supplementary Table 8). The key interactions are as follows: (1) the VP2 BC-loop residues T73 and E74 form hydrogen bonds with the heavy chain CDR1 (residue S30) and CDR2 (residue S52 and S53) loops; (2) the VP2 HI-loop residue K225 forms one hydrogen bond with heavy chain CDR1 residue T31, and also creates two salt bridges with heavy chain CDR3 residue D103; (3) the VP3 βI residue N211 forms a hydrogen bond with heavy chain CDR3 residue D105 (Fig. 6j). The 3H7 VL domain recognized two regions on the viral capsid that partially overlap with the 1F4 epitope, namely the VP3 knob (AB-loop) and VP1 C-terminus (Fig. 6k). However, these interactions are less extensive than those of 1F4 (Figs. 5j and 6k). The key interactions involve hydrogen bonds between: (1) the VP3 knob residue T61 and light chain CDR1 residue R32; (2) VP1 residue A286 with light chain CDR1 residue S31 (Fig. 6k). Notably, 3H7 maintains its binding mode in both CVA6 mature virions and expanded empty particles, with most interactions conserved and only some differing (Supplementary Fig. 14), indicating the 3H7 epitope remains accessible after capsid expansion.

To evaluate spatial relationships between the 3H7 epitope and KRM1 receptor-binding site, we compared the footprints of 3H7 and KRM1 on the CVA6 capsid using RIVEM analysis. The analysis shows that the 3H7 VL domain partially overlaps with the KRM1 binding site, whereas the VH domain does not (Fig. 6l). Structural superposition of the 3H7-CVA6-virion and KRM1-CVA6-virion complexes reveals that the 3H7 VL domain spatially blocks the KRM1 WSC domain from engaging the viral capsid (Fig. 6m). Together, these findings provide structural evidence that 3H7 can sterically block KRM1 from binding to CVA6, consistent with its functional blockade of KRM1 engagement (Fig. 2d).

### Comparison of the epitopes and binding kinetics of 1F4 and 3H7
Structural superposition of the 1F4-virion and 3H7-virion complexes shows that the two immune complexes adopt highly similar overall architectures. Both antibodies bind to the viral capsid near the three-fold axis and have overlapping binding footprints (Fig. 7a–c). However, a key difference is that 3H7's binding sites are closer to the three-fold axis than 1F4's, resulting in a more compact footprint and reduced steric interference with the adjacent KRM1 receptor-binding site (Fig. 7a–c). This structural observation appears contradictory to the

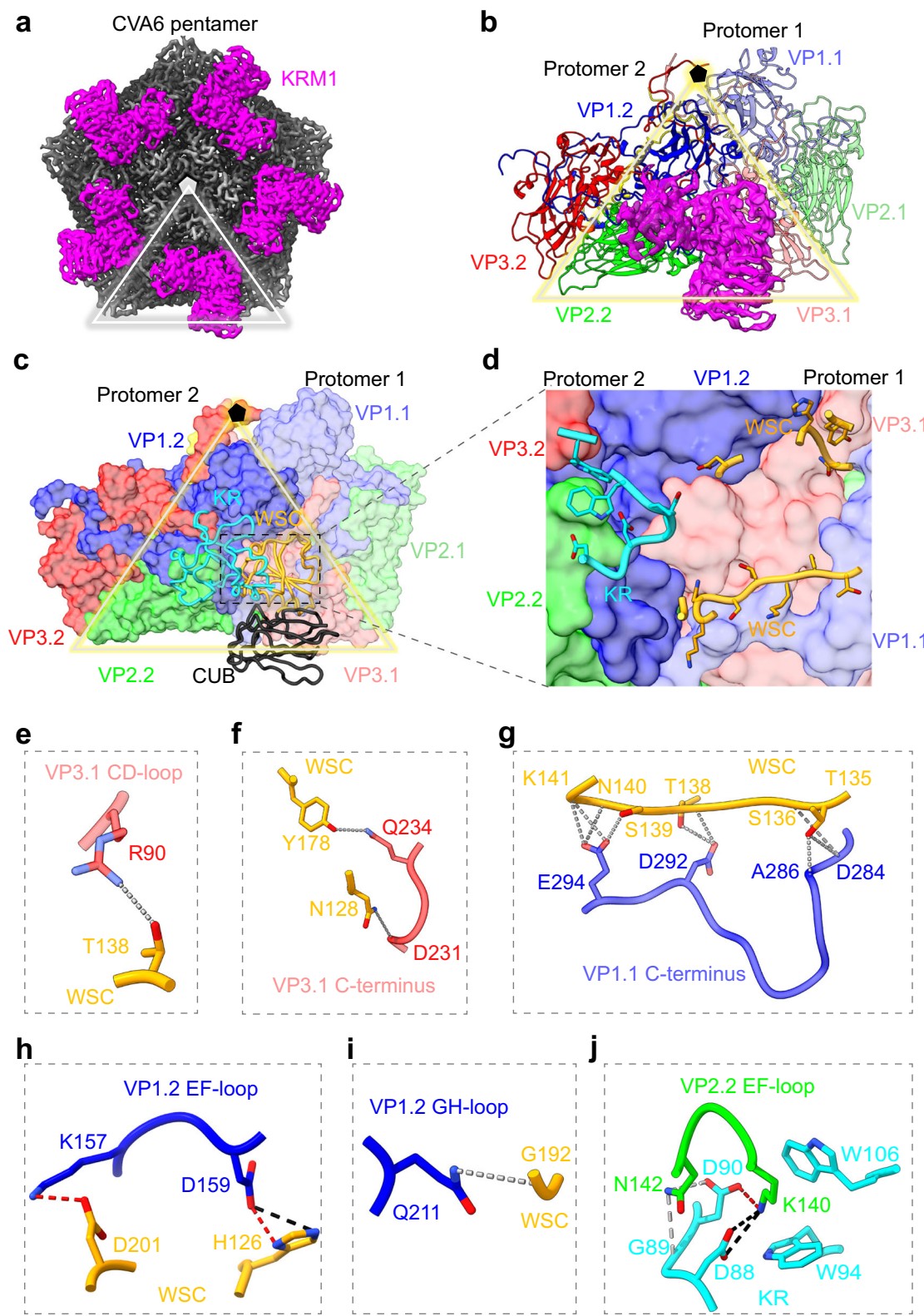

results from the KRM1 receptor competition ELISA assay, which shows that MAb 3H7 blocks KRM1 binding to CVA6 virion more effectively than MAb 1F4 (Fig. 2d). To address this discrepancy, we conducted a detailed comparison of the epitopes and binding affinities of the two antibodies.

Comparative epitope analysis reveals that the 3H7 antibody recognizes a broader array of structural elements, including the VP1 C-terminus, the VP2 BC and HI loops, and the VP3 AB loop and βI. In contrast, 1F4 focuses on a more restricted area, mainly consisting of the VP1 C-terminus and the VP3 AB-loop (Fig. 7c, d). Consistently, 3H7 exhibits a greater buried surface area (1124.4 Å²) than 1F4 (842.4 Å²) (Fig. 7d), suggesting stronger binding. In line with its broader interface, 3H7 exhibits superior virion-binding kinetics, with 8.6-fold higher affinity (KD = 0.40 nM vs. 1F4's KD = 3.43 nM) and 5-fold slower

**Fig. 4 | Molecular interactions between CVA6 and KRM1. a** Surface representation of a CVA6 capsid pentamer bound to KRM1, viewed along the five-fold symmetry axis. CVA6 pentamer is colored gray. KRM1 is magenta. One icosahedral asymmetric unit is marked by a triangle. Density maps displayed are unsharpened. **b** One KRM1 molecule interacts with two adjacent CVA6 protomers. KRM1 is shown as magenta ribbon, superimposed on cryo-EM density. CVA6 subunits are shown as ribbons and color-coded: Protomer 1 (VP1.1: light blue; VP2.1: light green; VP3.1: light red). Protomer 2 (VP1.2: blue; VP2.2: green; VP3.2: red). **c** Interaction of KRM1's KR and WSC domains with adjacent CVA6 protomers. KRM1 is drawn as a cartoon with the domains color-coded: KR, cyan; WSC, orange; CUB, black. CVA6 protomers are drawn as surface. **d** Enlarged view of the CVA6–KRM1 interaction interface, highlighting key residues on KRM1. **e–j** Binding interfaces between KRM1 and specific capsid regions: **e** VP3.1 CD-loop, **f** VP3.1 C-terminus, **g** VP1.1 C-terminus, **h** VP1.2 EF-loop, **i** VP1.2 GH-loop, **j** VP2.2 EF-loop. Hydrogen bonds are indicated by gray dashed lines; salt bridges by black dashed lines; regions with both hydrogen bonds and salt bridges are marked with red dashed lines.

dissociation rate (Kdis = $1.04 \times 10^{-4} s^{-1}$ vs. 1F4's Kdis = $2.08 \times 10^{-3} s^{-1}$) (Fig. 1e).

3H7's broader interface and superior binding characteristics should allow it to maintain more persistent viral particle occupancy than 1F4. To experimentally validate this, we conducted a BLI-based competitive binding assay. In this assay, immobilized CVA6-HeB virions were first incubated with either kinetics buffer (reference control) or the first antibody, followed by sequential exposure to the second antibody (Fig. 7e). The binding signals of secondary MAbs were analyzed, with results presented in Fig. 7f, g. The results confirmed asymmetric interference: when virions were pre-incubated with 3H7, subsequent 1F4 binding was completely blocked (Fig. 7f), whereas pre-bound 1F4 only partially inhibited later 3H7 binding (Fig. 7g). These findings indicate that the epitopes of 1F4 and 3H7 are overlapping but distinct, and also demonstrate that 3H7 has stronger retention on viral particles under competitive conditions.

The apparent contradiction is resolved by considering antibody binding kinetics. Although 3H7's compact binding creates less direct physical obstruction to KRM1, its broad epitope recognition and slow dissociation enable sustained occupancy, which effectively excludes receptor access. In contrast, 1F4's more extensive spatial interference with KRM1 is offset by its faster dissociation from the virion. These findings indicate that effective receptor blockade depends on both spatial blocking effects and binding kinetics rather than steric interference alone.

### Heparan sulfate mediates initial attachment of CVA6 to host cells followed by KRM1-dependent post-attachment entry

Both 1F4 and 3H7 MAbs block CVA6-KRM1 binding (Figs. 2d, 5k–l and 6l–m), but have little effect on viral attachment (Fig. 2b). This suggests KRM1 is not essential for CVA6's initial attachment. So, another receptor is likely involved in CVA6 attachment. Heparan sulfate proteoglycans (HSPG), widely expressed on host cell surfaces, serve as key attachment receptors for several enteroviruses, such as EV-A71 and CVA16[23,24]. However, whether HSPGs mediate attachment for CVA6 remains unknown. Structural analysis shows CVA6 virions have a conserved, positively charged patch at the five-fold axis (Fig. 8a), a region that mediates HSPG binding in EV-A71 and CVA16[23,24]. So, we hypothesized that HSPG may also help CVA6 attach to the cell surface. To test this, we systematically examined HSPG's role in CVA6 attachment and infection using genetic and biochemical methods.

Solute carrier family 35 member B2 (*SLC35B2*) encodes the 3′-phosphoadenosine 5′-phosphosulfate (PAPS) transporter 1 (PAPST1), a Golgi-localized transmembrane protein that transports PAPS from cytosol to Golgi for HSPG sulfation (Fig. 8b). To assess HSPG dependency in CVA6 infection, wild-type and CRISPR-engineered SLC35B2 knockout (*ΔSLC35B2*) RD cells[13] were infected with CVA6-HeB. Quantification of viral titers at 48 hpi showed a 35-fold reduction in *ΔSLC35B2* cells compared to wild-type controls, though less pronounced than the 132-fold decrease observed in *ΔKRM1* cells (Fig. 8c). These results demonstrate that CVA6 infection exhibits dual dependency, requiring both the KRM1 receptor and HSPG sulfation for productive infection.

To determine whether CVA6 directly interacts with heparan sulfate, CVA6-HeB culture supernatants were applied to a heparin-

immobilized agarose chromatography column. Following extensive PBS washes, bound virus particles were eluted with 2 M NaCl. Western blot analysis of column fractions using an anti-VP0 polyclonal antibody revealed striking enrichment of CVA6 capsid proteins (VP0 and VP2) in the high-salt eluate (Fig. 8d). In contrast, control experiment using empty agarose beads demonstrated no detectable binding of CVA6 particles (Fig. 8d). These results demonstrate that CVA6 virions specifically interact with heparin in vitro.

To assess the impact of heparin on CVA6 attachment, we pretreated CVA6-HeB with soluble heparin prior to its exposure to RD cells at 4 °C. Cell-bound viral RNA levels were quantified at 6 hpi by RT-qPCR. We found that heparin potently inhibited CVA6-HeB binding to RD cells in a concentration-dependent manner. Specifically, pretreatment with heparin at concentrations ≥500 μg/mL significantly inhibited viral attachment, whereas no inhibitory effect was observed at 50 μg/mL (Fig. 8e). These findings strongly suggest that HSPG act as attachment receptors for CVA6.

To directly compare the potential roles of KRM1 and HSPG in the CVA6 entry process, we evaluated their relative contributions to viral attachment and internalization. For attachment assays, pre-cooled wild-type RD, *ΔKRM1*, or *ΔSLC35B2* cells were incubated with CVA6-HeB at 4 °C to allow viral binding without internalization. Quantification of cell-associated viral RNA revealed no significant difference in attachment between *ΔKRM1* and wild-type cells (Fig. 8f), indicating that KRM1 is not essential for initial binding. In contrast, *ΔSLC35B2* cells exhibited an 82% reduction in viral attachment (Fig. 8f), demonstrating HSPG's dominant role in this step. For internalization assays, cell surface-bound virions (4 °C) were allowed to enter cells by shifting to 37 °C for 1 h, followed by trypsin treatment to remove residual surface virions. Quantification of internalized viral RNA revealed a 76% reduction in *ΔKRM1* cells and a 94% decrease in *ΔSLC35B2* cells compared to wild-type (Fig. 8g). Notably, *ΔKRM1* cells exhibited no attachment deficit but showed a marked 76% reduction in internalized virus (Fig. 8f, g), directly demonstrating that KRM1 is essential for post-attachment internalization independent of initial binding. The severe internalization defect observed in *ΔSLC35B2* cells (94% reduction compared to wild-type) likely results from two factors: impaired initial viral attachment (82% fewer surface-bound virions; Fig. 8f), which drastically reduces the pool of virus available for entry, and potential contributions of HSPG to post-attachment entry processes. Collectively, these data establish a two-receptor entry mechanism for CVA6: HSPG mediate the initial attachment of viral particles to the cell surface, while KRM1 facilitates their subsequent entry into the cytoplasm (Fig. 8h).

## Discussion

In this study, we systematically investigated the molecular mechanisms of CVA6 receptor recognition and antibody neutralization. We showed that purified CVA6-HeB primarily exists as mature virions. We identified KRM1 as the uncoating receptor for CVA6, as its binding induces A-particle formation. We propose a two-receptor entry model for CVA6: HSPG mediates attachment, and KRM1 facilitates uncoating. Additionally, we developed two protective MAbs, 1F4 and 3H7, which target a novel antigenic site near the viral capsid's three-fold axis. The

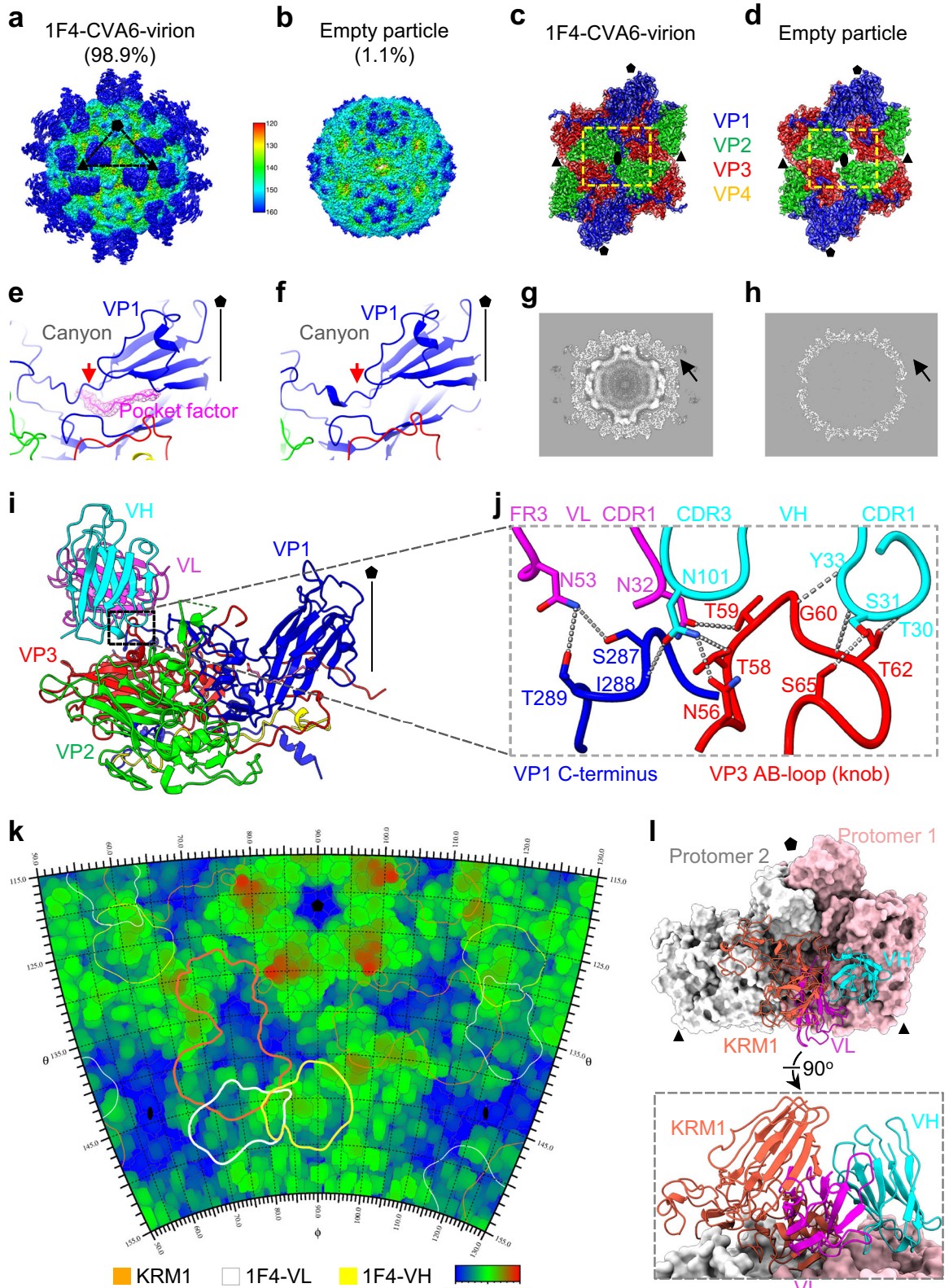

MAbs sterically block KRM1-virion interactions and function primarily at post-attachment steps, consistent with KRM1's uncoating role.

Research on viral structures is vital for developing effective vaccines and antiviral drugs. CVA6 is now the main global cause of HFMD. Yet, its structural characterization is limited, largely due to viral culturing difficulties[25]. Previously, only two CVA6 structural studies existed, analyzing the Gdula and 141 strains[8,9]. Both reported A-particles as

the predominant form, with mature virions being rare[8,9]. This led to the view that A-particles are CVA6's main infectious form and ideal vaccine target[9]. However, our study showed that purified CVA6-HeB particles are mostly mature virions, with no A-particles detected (Fig. 3). Thus, A-particle dominance isn't universal across all CVA6 strains. This discrepancy is likely attributable to a combination of strain-specific characteristics and methodological variations. Intrinsic differences

**Fig. 5 | Cryo-EM structures of CVA6 particles in complex with 1F4 Fab. a, b** Cryo-EM density maps of CVA6-HeB-1F4 in two conformations: 1F4-bound virion (**a**) and empty particle (**b**). Both maps are radially colored and viewed along the twofold axis. An icosahedral asymmetric unit is marked by a black triangle. 2D classification resolved 98.9% 1F4-bound virions and 1.1% empty particles. **c, d** Density maps of twofold-related protomers in the 1F4-bound virion (**c**) and empty particle (**d**), superimposed with atomic models. Conformational differences are outlined by yellow dashed rectangles. 1F4 Fab is omitted in (**c**) for clarity. **e, f** Atomic models of the VP1 hydrophobic pocket in the 1F4-bound virion (**e**) and empty particle (**f**). The pocket factor (magenta) is present in the 1F4-bound virion but absent in empty particle. Consequently, the VP1 pocket collapses in empty particle, accompanied by a significant inward shift of the VP1 GH loop (residues 223–225; red arrowhead,

RMSD = 1.4 Å). 1F4 Fab is omitted in (**e**) for clarity. The five-fold axis is indicated. **g, h** Central cross-sections of the 1F4-bound virion (**g**) and empty particle (**h**) along the z-axis. Black arrows indicate 1F4 Fab density. Central sections are displayed without masking or sharpening to preserve the native density information. **i** Atomic model of a CVA6 protomer bound to 1F4 Fab. VH is cyan and VL is magenta. The five-fold axis is indicated. **j** Enlarged views of CVA6–1F4 interaction interfaces. Hydrogen bonds are marked with gray dashed lines. **k** Surface footprint of 1F4 variable domains on the CVA6 virion using RIVEM analysis. Contour lines denote KRM1 (orange), 1F4 VL (white), and 1F4 VH (yellow). **l** The CVA6–1F4 complex structure was superimposed onto the CVA6–KRM1 complex, revealing spatial clashes between the 1F4 VL domain (magenta) and KRM1 (tomato). Protomer 1 is colored pink; Protomer 2 is colored white.

between isolates, such as capsid stability and susceptibility to uncoating, may account for the divergent particle composition. Furthermore, extrinsic factors including cell culture systems, purification protocols, and the criteria for selecting gradient fractions could also contribute to the observed variance. Future studies employing standardized methods to comparatively analyze these strains will be crucial to delineate the underlying causes. Notably, CVA6 virions bind efficiently to the KRM1 receptor, while A-particles bind poorly (Fig. 3). This indicates that mature virions are CVA6's main infectious form. In addition, the neutralizing antibody 1F4 binds effectively only to compact virions, not to expanded empty particles (Fig. 5), indicating its epitope is specific to the compact conformation. In summary, compact CVA6 virions offer more neutralizing epitopes than expanded particles, making them better candidates for vaccine development.

Enterovirus receptors are generally classified as attachment or uncoating receptors. KRM1 is the sole identified receptor for both CVA6 and CVA10 [15]. Previous research indicated that KRM1 acts as both the attachment and uncoating receptor for CVA10[15,17,18]. In our study, we discovered that KRM1 binds to mature CVA6 virions and triggers their conversion to A-particles (Fig. 3), suggesting its role in uncoating. However, *KRM1* knockout reduces viral infectivity without affecting attachment (Fig. 8). In contrast, disrupting HSPG by knocking out *SLC35B2* impairs both attachment and infectivity (Fig. 8). Moreover, CVA6 virions directly bind to heparin-beads, and soluble heparin can potently inhibit CVA6 binding to RD cells (Fig. 8). These results indicate that HSPG, rather than KRM1, serves as the primary attachment receptor for CVA6. Additionally, studies on neutralizing antibodies, such as 1F4, have shown that while the antibody can sterically block KRM1-virion interactions, it does not prevent viral attachment (Fig. 2, Fig. 5), further indicating that KRM1 functions in post-attachment steps. These findings support a two-receptor entry mechanism for CVA6: HSPG mediates viral attachment, while KRM1 facilitates viral uncoating. This mechanism is analogous to that of EV-A71, where HSPG mediates attachment and SCARB2 acts as the uncoating receptor[12,13].

Neutralizing MAbs are promising antiviral candidates. The only previously reported CVA6-specific neutralizing MAb, 1D5, targets the five-fold vertex and blocks viral attachment[9]. In this study, we developed two novel CVA6-specific neutralizing MAbs, 1F4 and 3H7, which showed excellent preventive and therapeutic efficacy against lethal CVA6 infection in mice (Fig. 1, Supplementary Fig. 1). Cryo-EM studies revealed that both MAbs bind to a novel antigenic site near the viral capsid's three-fold axis. This site includes the VP1 C-terminus, VP2 BC and HI loops, and VP3 AB loop and βI (Figs. 5–7). Notably, the VP1 residues A286 to T289 in this site are adjacent to the VP1 C-terminal peptide P59 (residues 291-305), a known linear B-cell epitope recognized by CVA6 virus-like particle (VLP) immune sera[26]. Furthermore, this antigenic site corresponds to EV-D68 site III, which consists of EV-D68 VP1 C-terminal residues 285 and 293[27]. Biochemical and structural analyses show that MAbs 1F4 and 3H7 sterically block KRM1 receptor binding and function primarily at post-

attachment steps (Fig. 2 and 5–6). Thus, our study reveals a novel CVA6 antibody epitope and a distinct neutralization mechanism.

Our data demonstrate that MAbs 1F4 and 3H7 are highly specific for CVA6. Neutralization assays confirmed that neither antibody exhibited cross-neutralizing activity against a panel of related viruses, including CVA10 (Fig. 1a) and other KRM1-using enteroviruses such as CVA2, CVA3, CVA4, CVA5, CVA8, and CVA12, even at the highest concentration of 10 μg/mL (Supplementary Fig. 15a). The molecular basis for this high specificity is revealed by our sequence analysis, which shows that the key capsid residues constituting the 1F4 and 3H7 epitopes are not conserved among other KRM1-using enteroviruses (Supplementary Fig. 15b). While this indicates that 1F4 and 3H7 are not broad-spectrum therapeutics, this high specificity is a crucial finding. It definitively shows that the epitopes targeted by our antibodies are unique to CVA6, making them highly valuable as precise tools for CVA6-specific diagnosis and pathogenesis studies.

Collectively, our findings establish a two-receptor entry mechanism for CVA6, with HSPG mediating attachment and KRM1 facilitating uncoating. We also reveal the structural basis for therapeutic antibody intervention. This study enhances our understanding of CVA6 infection and pathogenesis, and offers key insights for developing anti-CVA6 vaccines and drugs.

## Methods
### Cells and viruses
Human rhabdomyosarcoma (RD) wild-type and knockout cells were cultured in DMEM (Gibco, Thermo Fisher Scientific, USA) supplemented with 10% fetal bovine serum (FBS) at 37 °C under standard culture conditions.

CVA6 strain 54203/HeB/CHN/2012 (HeB; GenBank ID: MK106189)[20] and CVA10 strain S0148b (GenBank ID: KX094564) were propagated in RD cells. CVA6 strain TW-2007-00141 (141; GenBank ID: KR706309) was successfully rescued from infectious clone[28]. Viral titers were quantified by 50% tissue culture infectious dose ($TCID_{50}$) assays using RD cells.

### Proteins and antibodies
CVA6 virions were purified from RD cells infected with CVA6-HeB or CVA6-141. At 72–96 hpi, supernatants and cell lysates were clarified via high-speed centrifugation and then precipitated with 10% PEG 8000 and 200 mM NaCl. Pelleted virions were resuspended in 0.15 M PBS, purified sequentially through 20% sucrose cushion at 112,700 × g for 5 h and 10–50% sucrose gradients at 270,000 × g for 3 h. Virus-containing fractions were confirmed by SDS-PAGE and quantified by Bradford assay.

The His-tagged hKRM1-Fc fusion protein, comprising the human KRM1 ectodomain (residues A23–G373) fused to the human IgG1 Fc region, was engineered and purified from HEK293F cells via nickel affinity chromatography[29]. Similarly, the His-tagged hKRM1 ectodomain (residues A23–G373) was also prepared following the same protocol.

Polyclonal antibodies against CVA6 VP0 were generated by immunizing BALB/c mice with recombinant CVA6 VP0 protein

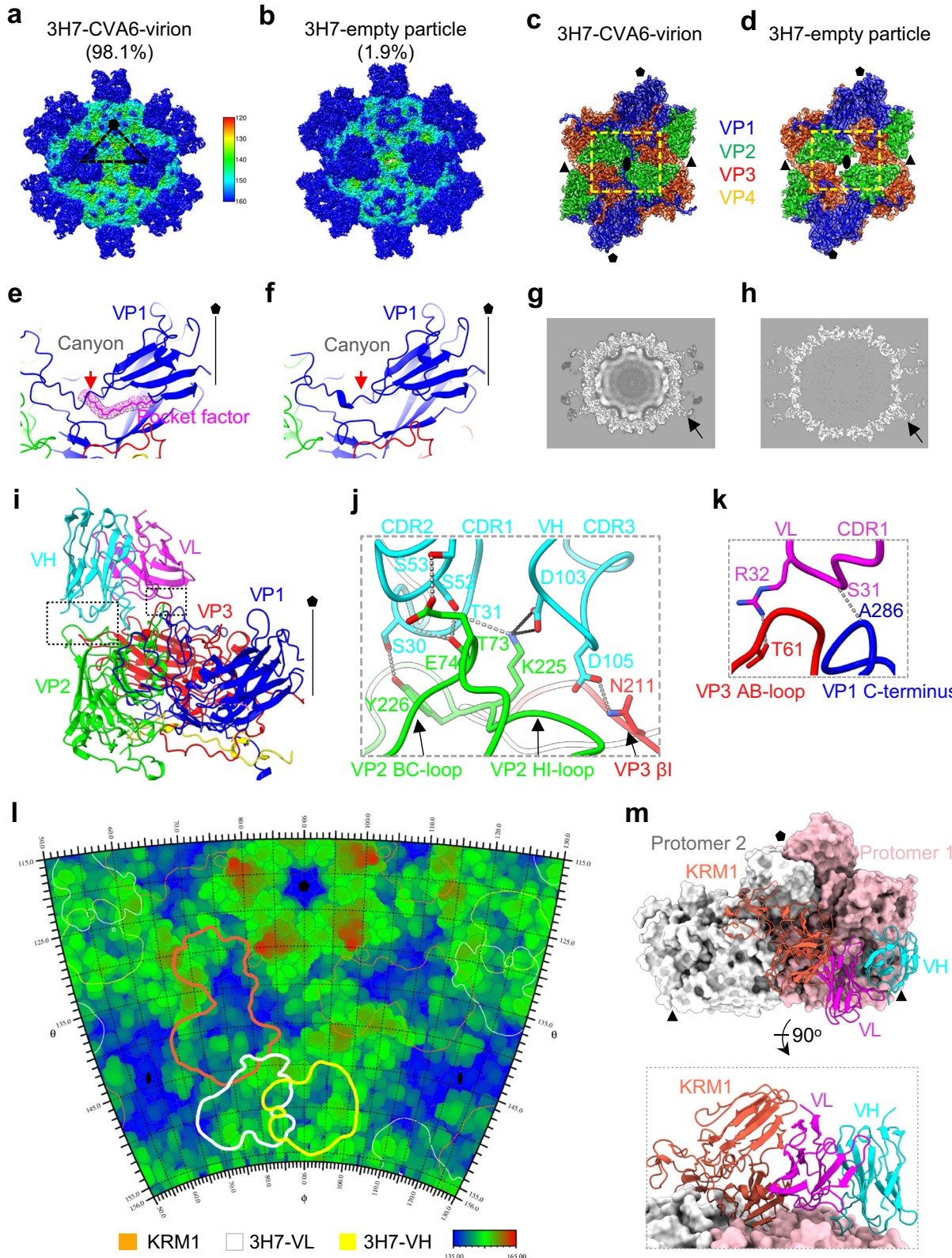

**a** 3H7-CVA6-virion (98.1%)  **b** 3H7-empty particle (1.9%)  **c** 3H7-CVA6-virion  **d** 3H7-empty particle

VP1, VP2, VP3, VP4

**e** VP1 Canyon, Pocket factor  **f** VP1 Canyon  **g**  **h**

**i** VH, VL, VP3, VP1, VP2  **j** CDR2, CDR1, VH, CDR3; S53, S52, D103, T31, S30, T73, E74, K225, D105, N211, Y226, VP2 BC-loop, VP2 HI-loop, VP3 βI  **k** VL, CDR1, R32, S31, A286, T61, VP3 AB-loop, VP1 C-terminus

**l** KRM1 | 3H7-VL | 3H7-VH; 135.00–165.00

**m** Protomer 2, KRM1, Protomer 1, VH, VL; 90°; KRM1, VL, VH

expressed in *Escherichia coli*, emulsified in Freund's adjuvants[22]. MAb 3A2, an IgG antibody targeting SARS-CoV-2[19], served as a negative control in this study.

## Preparation and sequencing of anti-CVA6 MAbs

To generate anti-CVA6 MAbs, adult female BALB/c mice were immunized intraperitoneally three times with 5 μg/dose of purified CVA6-141

virions in aluminum adjuvant at 2-week intervals. About two weeks after the final immunization, one mouse was intravenously boosted with 20 μg of purified CVA6-141 virions. Splenocytes were harvested three days post-boost and fused with SP2/0 myeloma cells using PEG 1450 (Sigma, USA), followed by HAT selection. Hybridoma supernatants were screened via CVA6-141 neutralization assays (described below), and positive clones were subcloned 2–4 times to ensure

**Fig. 6 | Cryo-EM analysis of CVA6–3H7 interactions. a, b** Cryo-EM density maps of CVA6-HeB-3H7 resolved into two states: **a** 3H7-bound virion (98.1%) and **b** 3H7-associated empty particle (1.9%). **c, d** Structural features of the twofold axis channel in the 3H7-bound virion (**c**) and 3H7-associated empty particle (**d**). Structural differences are outlined by yellow dashed rectangles. 3H7 Fab is removed for clarity. **e, f** Structures of VP1 hydrophobic pocket in the 3H7-bound virion (**e**) and 3H7-associated empty particle (**f**). A pocket factor (magenta) is present in the virion but absent in the empty particle, leading to pocket collapse and an inward displacement of the VP1 GH loop (red arrowhead). 3H7 Fab is omitted in both panels for clarity. The fivefold axis is indicated. **g, h** Central cross-sections of the 3H7-bound

virion (**g**) and 3H7-associated empty particle (**h**). Black arrows indicate 3H7 Fab density. **i** Atomic model of a CVA6 protomer in complex with 3H7 Fab. VH is cyan, and VL is magenta. The fivefold axis is labeled. **j, k** Enlarged views of interaction interfaces between CVA6 and the 3H7 VH (**j**) or VL (**k**). Gray dashed lines represent hydrogen bonds. Black dashed lines indicate salt bridges. **l** Footprint analysis of 3H7 variable domains on CVA6 virion. Contour lines mark KRM1 in orange, 3H7 VL in white, and 3H7 VH in yellow. **m** Structural superposition of the CVA6–3H7 and CVA6–KRM1 complexes, demonstrating steric clashes between 3H7 VL (magenta) and KRM1 (tomato). Protomer 1 is pink, and protomer 2 is white.

monoclonality. Antibody isotypes were determined using an HRP-based ELISA kit (Southern Biotech, USA). Variable region sequences of heavy and light chains were amplified with mouse Ig primers (Novagen, Germany) and analyzed via IgBLAST. MAbs were purified from ascites using protein G agarose (Yeasen, China).

### Neutralization assay
Undiluted hybridoma supernatants or serially diluted MAbs (50 μl/well) were mixed with 100 TCID$_{50}$ of CVA6 or CVA10 in 96-well plates and incubated at 37 °C for 1 h. RD cells (20,000 cells/well) were added, and plates were incubated for ~3 days. CPE was visually assessed, and cell viability was quantified using CellCounting-Lite® 2.0 kit (Vazyme). Percent neutralization was calculated as: 100 × (RLU of sample - RLU of virus control) / (RLU of untreated cells - RLU of virus control). IC50 values were determined via nonlinear regression (GraphPad Prism).

### Antibody binding ELISA
ELISA plates were coated overnight at 4 °C with 100 ng/well of purified CVA6-HeB virions and then blocked with 5% milk. After washes, two-fold serially diluted MAbs was added (50 μL/well) and incubated for 2 h at room temperature. After washing, HRP-conjugated anti-mouse IgG (1: 10,000; Proteintech, China) was added and incubated for 1 h at room temperature. Absorbance at 450 nm was measured after TMB color development.

### Bio-layer interferometry (BLI) assay
CVA6-HeB virions were biotinylated using EZ-Link Sulfo-NHS-LC-LC-Biotin (Thermo Fisher) and purified via Zeba spin desalting columns. Virus-antibody binding affinity was analyzed on an Octet RED96 system (Pall FortéBio). Briefly, biotinylated CVA6 virions were immobilized on streptavidin biosensors, exposed to serially diluted MAbs for association (500 s), and then transferred to dissociation buffer (PBS with 0.1% BSA/0.02% Tween 20) for 500 s. Equilibrium dissociation constants (KD) were calculated using Octet software.

For competition assays, CVA6 virion-coated sensors were sequentially incubated with buffer (control) or 15 μg/mL first MAb for 500 s, followed by 15 μg/mL second MAb alone or combined with first MAb (to prevent dissociation of pre-bound molecules) for 500 s. Binding levels of the second MAb were quantified via Octet software.

### In vivo protection assays
The protective efficacy of MAbs 1F4 and 3H7 was assessed in neonatal ICR mouse models infected with highly lethal CVA6 strains HeB[20] and S0087b[21,22]. Note that CVA6-S0087b lacks detectable CPE in vitro. To standardize challenge doses, the 50% lethal doses (LD50) of CVA6-HeB and CVA6-S0087b were determined in neonatal ICR mice.

For prophylactic evaluation, groups of one-day-old ICR mice received intraperitoneal (i.p.) injections of PBS, 10 mg/kg of anti-CVA6 MAbs or control antibody[19], followed 24 h later by i.p. challenge with ~10 LD50 of CVA6-HeB or ~25 LD50 of CVA6-S0087b.

For therapeutic evaluation, groups of two-day-old ICR mice were i.p. infected with ~10 LD50 of CVA6-HeB or ~25 LD50 of CVA6-S0087b. After 24 h, mice received i.p. injections of PBS or 10 mg/kg MAbs. For

both assays, all infected mice were monitored for 14 days to record survival and clinical symptoms.

### Time-of-addition assay
The assay was performed according to our previously described protocol[30,31]. Briefly, for pre-attachment inhibition, 1000 TCID$_{50}$ of CVA6-HeB was mixed with 1 μg of MAb 1F4 or 3H7 for 1 h, cooled on ice, and applied to pre-chilled RD cells in 24-well plates for viral attachment at 4 °C for 2 h. The cells were washed twice with ice-cold PBS and incubated in DMEM with 1% FBS at 37 °C.

For post-attachment inhibition, 1000 TCID$_{50}$ of CVA6-HeB was adsorbed to pre-chilled RD cells at 4 °C for 2 h. After cold PBS washes, cells were incubated at 37 °C for 0 or 0.5 h to initiate viral entry, followed by treatment with 1 μg of MAb 1F4 or 3H7 in fresh medium at 37 °C.

In both assays, total RNA was extracted 6 h post-infection using VeZol reagent (Vazyme). cDNA was synthesized using the HiScript III 1st Strand cDNA Synthesis Kit (Vazyme). Quantitative PCR was performed with SYBR Premix Ex Taq (Takara) to measure CVA6 RNA levels, which were normalized to the housekeeping gene β-actin. Specific primers for CVA6 (Forward: 5′-TACTTTGGGTGTCCGTGTTT-3′, Reverse: 5′-TGGCCAATCCAATAGCTATATG-3′)[9] and β-actin (Forward: 5′-GGACTTCGAGCAAGAGATGG-3′, Reverse: 5′-AGCACTGTGTTGGCGTACAG-3′) were used.

### Inhibition of virus attachment by the MAbs
CVA6-HeB (50,000 TCID$_{50}$) was pre-incubated with MAbs 1F4, 3H7, or control antibody (1, 10, or 100 ng) at 37 °C for 1 h. The mixtures were ice-cooled and added to pre-chilled RD cells in 24-well plates for 2 h adsorption at 4 °C. After ice-cold PBS washes, cell-associated viral RNA was extracted with VeZol reagent (Vazyme). cDNA synthesis and qPCR were performed as described above, with CVA6 RNA levels normalized to β-actin.

### Receptor blockade ELISA
ELISA plates were coated with purified CVA6-HeB virions (50 ng/well) overnight at 4 °C and blocked with 5% non-fat milk in PBST for 1 h at room temperature. Serially diluted anti-CVA6 MAbs or control antibody were mixed with 12.5 ng/well of biotinylated hKRM1-hFc[29] and transferred to the virus-coated plates. After 1 h incubation at room temperature and washing, bound biotinylated hKRM1-hFc was detected using HRP-conjugated streptavidin (1:5000; Proteintech) for 1 h. Absorbance at 450 nm was measured following TMB substrate development.

### Fab preparation
IgG antibodies 1F4 and 3H7 were buffer-exchanged into sample buffer (20 mM phosphate, 10 mM EDTA, pH 7.0) using ultrafiltration. The antibodies were digested with papain-agarose in PBS containing 2 mM TCEP (pH 7.0) at 37 °C with rotation. Digestion progress was monitored by periodic SDS-PAGE analysis of aliquots. After complete digestion, samples were purified using Q-column chromatography (Cytiva) to collect Fab fragments in the flow-through. These Fabs were then purified by size-exclusion chromatography (Superdex 200

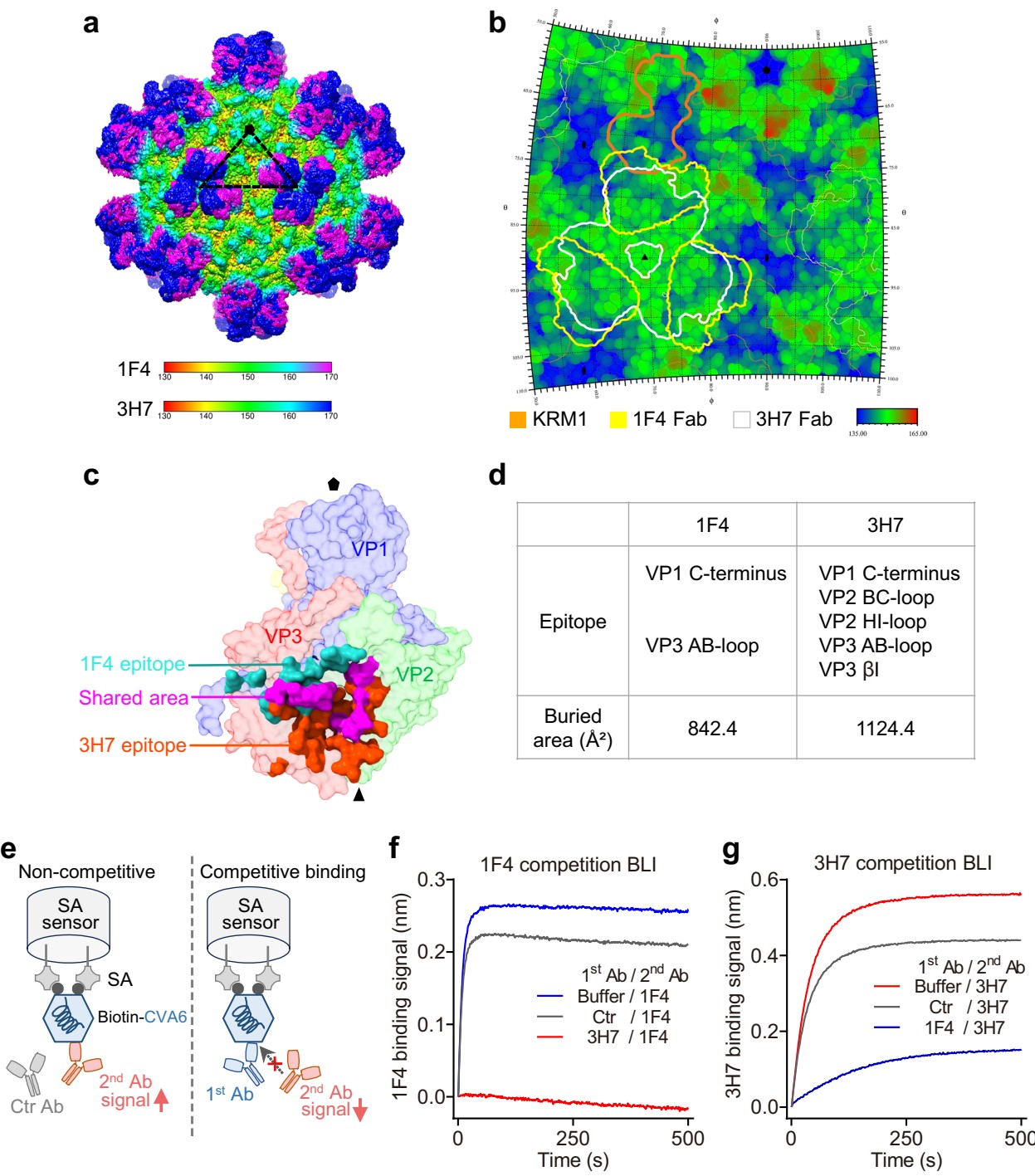

**Fig. 7 | Competitive binding of 1F4 and 3H7 to overlapping epitopes on CVA6.**
**a** Superimposed cryo-EM density maps of CVA6-1F4 and CVA6-3H7 complexes. The maps are colored by radial distance. 1F4 Fab is shown in magenta. 3H7 Fab is shown in blue. **b** Footprint analysis of 1F4 and 3H7 on CVA6 surface. Contour lines mark KRM1 in orange, 1F4 Fab in yellow, and 3H7 Fab in white. **c** Surface representation of CVA6 protomer showing epitopes of 1F4 (light sea green) and 3H7 (orange red). Overlapping regions recognized by both antibodies are highlighted in magenta. **d** Summary of the epitopes and buried surface areas of the 1F4 and 3H7 MAbs. **e** Schematic of antibody competition assay by BLI. (Left) No competition: the

second CVA6 MAb binds freely to immobilized virions, producing strong signal. (Right) Competition: the first antibody binds to overlapping epitope, blocking the second MAb binding and reducing signal. **f**, **g** BLI-based antibody competition assay. Immobilized CVA6-HeB virions were pre-incubated with buffer (reference) or the first antibody, then exposed to the second MAb 1F4 (**f**) or 3H7 (**g**). An anti-SARS-CoV-2 MAb served as control (gray curve). Sensorgrams show real-time binding signals of the second MAbs 1F4 (**f**) and 3H7 (**g**). Source data are provided as a Source Data file.

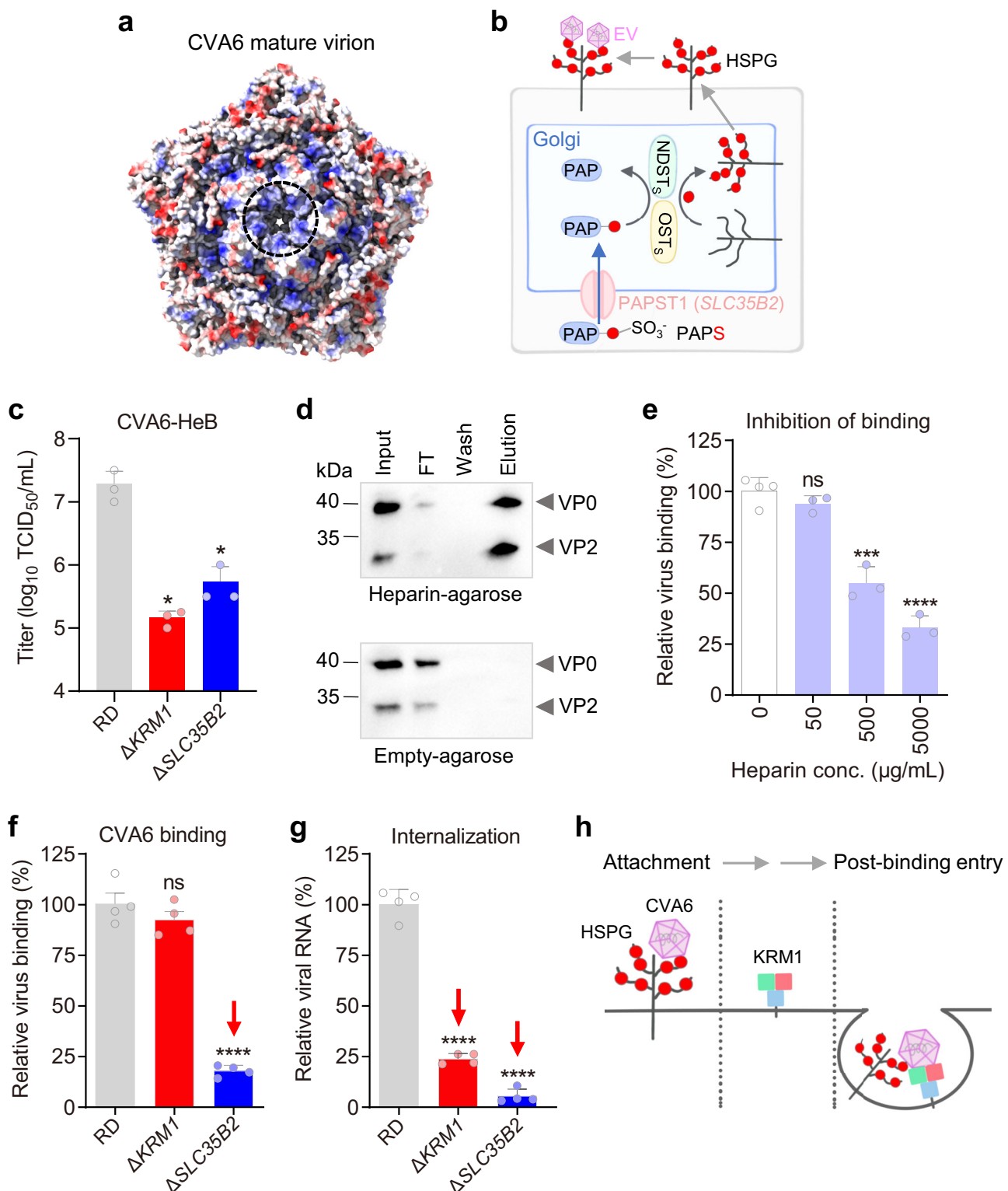

Increase, Cytiva) and quantified by $A_{280}$ absorbance (Nanodrop, Thermo).

## Cryo-EM sample preparation and data collection

Purified CVA6-HeB (0.11 mg/mL) was incubated with excess KRM1 or Fab (molar ratio ~1:120) in PBS (pH 7.4) for 30–35 min at room temperature. Then, 3 µL of apo-CVA6 or CVA6 complexes with KRM1/Fab were applied to glow-discharged 200-mesh lacey carbon grids (Ted Pella, USA). Grids were blotted and vitrified by plunging into liquid ethane using a Vitrobot Mark IV (Thermo, USA) at 4 °C and 100% humidity.

Apo-CVA6 datasets were acquired on a Thermo Fisher Titan Krios (300 kV) equipped with a Falcon IV detector using EPU software. Micrographs of KRM1-CVA6 and Fab-CVA6 complexes were recorded in super-resolution mode on a JEOL cryoARM 300 electron microscope equipped with a K3 direct electron detector (Gatan). Detailed data collection parameters are provided in Supplementary Tables 1, 2, 4, 7.

**Fig. 8 | Heparan sulfate serves as the primary attachment receptor for CVA6, whereas KRM1 functions as a post-attachment entry receptor. a** Electrostatic surface potential maps of CVA6-HeB viral pentamers. Color scheme: red for negative potential, white for neutral, and blue for positive potential. The black dashed circle highlights the positively charged patch. **b** Schematic diagram illustrating the sulfation process of heparan sulfate and its role in mediating cellular attachment of multiple enteroviruses (EV). **c** Wild-type RD cells, $\Delta KRM1$ cells, and $\Delta SLC35B2$ cells were infected with CVA6-HeB, and viral titers were quantified at 48 hpi. Data are means ± SD of triplicate biological samples. $p = 0.0422$ ($\Delta KRM1$), $p = 0.045$ ($\Delta SLC35B2$). **d** CVA6 specifically binds heparin-agarose beads. CVA6-HeB supernatant was loaded onto heparin or control columns, followed by washing and elution. Samples were analyzed by western blotting with anti-CVA6-VP0 antibody. FT, flow through. Two independent experiments were performed, with similar results. **e** Inhibition of CVA6 attachment to RD cells by soluble heparin. CVA6-HeB

was treated with heparin and allowed to attach to cells for 2 h at 4 °C. Attached virus was quantified by RT-qPCR analysis and normalized to β-actin. Data are means ± SD (n = 4 for virus-only; n = 3 for others). $p = 0.2084$ (heparin-50 μg/ml), $p = 0.0004$ (heparin-500 μg/ml). **f** Viral attachment assay. CVA6-HeB was incubated with wild-type RD cells, $\Delta KRM1$ cells, and $\Delta SLC35B2$ cells at 4 °C for 2 h. After washing, cell-bound virus was quantified via RT-qPCR. Data are means ± SD of four biological replicates. $p = 0.2859$ ($\Delta KRM1$). **g** Viral internalization assay. After virus binding at 4 °C, cells were shifted to 37 °C. Internalized virus was quantified by RT-qPCR following trypsin removal of surface virions. Data are means ± SD of four biological replicates. Source data are provided as a Source Data file. Statistical note for (**c, e, f, g**): significance was determined by two-tailed t-test. ns, $p \geq 0.05$; *, $p < 0.05$; **, $p < 0.01$; ***, $p < 0.001$; ****, $p < 0.0001$. **h** Two-step model of CVA6 entry: attachment via heparan sulfate, followed by KRM1-mediated internalization.

## Structure determination

Micrographs underwent patch-based motion correction and CTF estimation in cryoSPARC (version 4.5.3)[32], followed by particle picking targeting ~30 nm viral particles. After 2D classification, particle classes showing distinct morphologies were selected for separate model building. Initial models were generated via ab initio reconstruction in cryoSPARC or using previously reported CVA6 density maps (Supplementary Figs. 3, 6, 11, and 12). Subsequent iterative heterogeneous, homogeneous, or non-uniform refinement was performed, and the resulting density maps were used for model building, refinement, validation, and structural interpretation. The maps were further sharpened using DeepEMhancer to facilitate model refinement[33]. Atomic models of viral protomers, Fabs, and KRM1 were predicted using AlphaFold3[34] and manually fitted into density maps using UCSF Chimera (version 1.18)[35]. Models were refined in WinCoot (version 0.9.8.95) and PHENIX (version 1.17.1)[36], and validated using MolProbity (in Phenix) and the PDBe Validation Server. Final structures were deposited in PDB. Interaction interfaces were analyzed using PDBePISA[37]. All structural figures were prepared using UCSF Chimera and ChimeraX (version 1.9)[38].

## Virus infection assay

RD wild-type, $\Delta KRM1$[16], and $\Delta SLC35B2$[13] cells in 24-well plates were infected with CVA6-HeB at an MOI of 0.15 for 1 h at 37 °C. After PBS washing, cells were incubated in fresh medium at 37 °C for 24 h or 48 h. Cells and culture supernatants were harvested, subjected to freeze-thaw cycles, and viral titers were quantified using the TCID$_{50}$ method.

## KRM1 receptor-binding ELISA

To assess KRM1 receptor-binding activity of CVA6, ELISA plates were coated overnight at 4 °C with serially diluted purified CVA6-HeB virions. After blocking with 5% milk, the plates were incubated with 100 ng/well of hKRM1-Fc or ACE2-Fc (control protein)[19] for 1 h at room temperature. Following washes, horseradish peroxidase (HRP)-conjugated anti-human IgG (Proteintech) was added and incubated for 1 h at room temperature. Absorbance was measured at 450 nm after color development.

## Binding of CVA6 to heparin-agarose beads

Chromatography columns were packed with heparin-agarose (Yeasen, China) or empty agarose beads (control) and equilibrated with 0.01 M PBS. CVA6-HeB culture supernatant (5 mL, serum-free) was applied to the columns. After loading, columns were washed with 10 column volumes of PBS. Bound viral particles were eluted with elution buffer (0.01 M PBS, 2 M NaCl). Each fraction was analyzed for the presence of CVA6 proteins by western blotting using an anti-VP0 polyclonal antibody as primary antibody (1:1000 dilution) and goat anti-mouse IgG–HRP (Proteintech) as secondary antibody (1:10,000 dilution).

## Inhibition of CVA6 attachment with soluble heparin

To evaluate the inhibitory effect of heparin on CVA6 attachment to RD cells, CVA6-HeB (1000 TCID$_{50}$) was mixed with serially diluted heparin sodium salt (Sanjie, Shanghai, China) and incubated at 37 °C for 1 h to allow potential interactions between heparin and viral particles. The mixture was then cooled and added to precooled RD cells in a 24-well plate, followed by incubation at 4 °C for 2 h under conditions that permitted viral attachment. After incubation, unbound virus was removed by washing the cells twice with ice-cold PBS. Cell-bound viral RNA was extracted using VeZol reagent (Vazyme, China). cDNA synthesis and qPCR were performed as described above, with CVA6 RNA levels normalized to β-actin.

## CVA6 attachment assay

The CVA6 attachment assay was conducted by adding CVA6-HeB (50,000 TCID$_{50}$) to precooled RD wild-type, $\Delta KRM1$[16], or $\Delta SLC35B2$[13] cells in 24-well plates, followed by incubation at 4 °C for 2 h to allow viral attachment. Unbound virus was removed with ice-cold PBS washes, and cell-bound viral RNA was extracted using VeZol reagent (Vazyme). cDNA synthesis and qPCR were performed as described above, with CVA6 RNA levels normalized to β-actin.

## CVA6 internalization assay

The CVA6 internalization assay was performed by first binding CVA6-HeB to pre-cooled RD wild-type, $\Delta KRM1$[16], or $\Delta SLC35B2$[13] cells in 24-well plates at 4 °C for 2 h. After washing, cells were transferred to 37 °C for 1 h to permit viral internalization. Surface-bound virions were removed by trypsin treatment and additional washes. RNA extraction, cDNA synthesis, and qPCR were carried out as described above, with CVA6 RNA levels normalized to β-actin to quantify internalized virus.

## Ethics statement

The mouse studies were approved by the Institutional Animal Care and Use Committee of Fudan University. Mice were purchased from Vital River Laboratory Animal Technology company.

## Statistical analysis

All statistical analyses were performed using GraphPad Prism version 8.

## Reporting summary

Further information on research design is available in the Nature Portfolio Reporting Summary linked to this article.

# Data availability

The atomic coordinates of the CVA6 mature virion, CVA6 empty particles, KRM1-CVA6-virion complex, KRM1-triggered CVA6 A-particle, 1F4-CVA6-virion complex, 3H7-CVA6-virion complex, and 3H7-CVA6 empty particle complex have been deposited in the Protein Data Bank (PDB) under accession codes 9VFQ, 9VFP, 9VFR, 9VFS, 9VFU, 9VFT and

9VG1, respectively. The corresponding unsharpened cryo-EM density maps have been deposited in the Electron Microscopy Data Bank (EMDB) under accession codes EMD-65031, EMD-65030, EMD-65032, EMD-65034, EMD-65038, EMD-65036 and EMD-65043, respectively. The sequences of 1F4-VH, 1F4-VL, 3H7-VH, and 3H7-VL have been deposited in the DNA Data Bank of Japan (DDBJ) under accession codes LC900916, LC900917, LC900918, and LC901458, respectively. Source data are provided with this paper.

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

## Acknowledgements

We thank Wei Wei (Jilin University) and Peng Gong (Wuhan Institute of Virology) for helpful discussions. We thank Dr. Yong Zhang (Chinese Center for Disease Control and Prevention) for the CVA6-HeB strain, Dr. Tong Cheng (Xiamen University) for the CVA6-141 infectious clone plasmid, Dr. Mingzhou Chen (Hubei University) for the ΔSLC35B2 RD cells, and Dr. Shuye Zhang (Fudan University) for the ΔKRM1 RD cells. We thank Dr. Yue Liu (Zhejiang University) for the instruction of RIVEM. We also thank Dr. Guibo Rao and Dr. Mingyu Wei (Center for Instrumental Analysis and Metrology, Wuhan Institute of Virology) and the Center of Advanced Analysis & Gene Sequencing at Zhengzhou University for their technical support. C.Z. received grants from the National Natural

Science Foundation of China (32170948 and 82472252), National High-Level Talent Special Support Programs (10,000 Talents Program) - Young Talents, and the Shanghai Municipal Science and Technology Major Project (ZD2021CY001). B.S. received a grant from the National Key Research and Development Program (2025YFC2608600). This study was also supported by a grant from the National Natural Science Foundation of China (U23A20147) to Peng Gong. W.L. was supported by a grant from the National Natural Science Foundation of China (32470163). The funders played no role in study design, data collection, analysis, manuscript preparation, or publication decisions.

## Author contributions

C.Z., B.S., and X.K. designed the experiments. X.L., Z.L., K.L., and X.Y. performed the biochemical and animal experiments. X.K. collected the cryo-EM data and performed cryo-EM reconstructions and model building with the help of W.L. C.Z., X.K., and B.S. analyzed the data. C.Z., X.K., and B.S. wrote the manuscript.

## Competing interests

C.Z., X.L., and Z.L. are inventors on a pending patent for anti-CVA6 MAb 3H7. All other authors declare no competing interests.
