## [Peer Review File · Nature Communications]

Molecular mechanisms of receptor recognition and antibody neutralization of coxsackievirus A6

Corresponding Author: Dr Chao Zhang

Version 0:

Reviewer comments:

Reviewer #1

(Remarks to the Author)

Referee report:

Molecular mechanisms of receptor recognition and antibody neutralization of coxsackievirus A6

In the offered manuscript Ke et al describe a 2 receptor entry mechanism for CVA6. CVA's are a significant concern for human health and a deeper understanding of how these viruses infect humans could lead to the development of prophylactics. The authors further develop and describe the mode of action of two MABs of potential use to treat CVA6 infections. Overall this reviewer found the study to be soundly conducted and does not have any major criticisms to offer.

At most, this reviewer is curious if the authors could comment on whether their antibodies can also neutralise related KRM1 binding CVs? Having a broad-spectrum therapeutic would raise the overall impact of the story.

Minor comments:

Line 61: which stabilizes the capsid.
Requires citation.

Line 694: which were sharpened with DeepEMhancer

Can the authors comment for what purpose the DeepEMhancer maps were used? While it is acceptable to use modified maps to aid model building, the final model should be refined against the unsharpened maps. Moreover, based on the deposition codes, it isn't clear if the authors have deposited just the sharpened maps or if they have also deposited the unsharpened half maps. This reviewer encourages that all maps that were used in the interpretation of the densities be deposited (eg, unsharpened averaged map, globally sharpened, map filtered by local resolution (if it exists and was used), and deepEMhanced maps).

Figure 3(R-T), Line 966 and Figure 5(G-H), Lines 1001-1002: Please could the authors state if the central sections through the maps are masked and sharpened or not.

Figure 4A-B: Are the represented densities from sharpened maps or deepEmHanced ones?

Figure 5 and 6 (E-F): The cartoon representations could do with additional captioning to aid readers. Eg. indicate where the 5 fold axis is, highlighting re-organisation of the loops involved in the binding of the pocket factor.

Reviewer #2

(Remarks to the Author)

The manuscript "Molecular mechanisms of receptor recognition and antibody neutralization of coxsackievirus A6" by Xianliang Ke et al details the likely host entry mechanisms of CVA6 involving both heparan sulfate proteoglycan (HSPG) for attachment and receptor Kringle-containing transmembrane protein 1 (KRM1) to trigger uncoating. This 2-step process is unlike another coxsackievirus that uses KRM1 for both and is highly relevant for developing antivirals given the recently

recognized role of CVA6 as a leading causative agent of hand, foot, and mouth disease. Previous structural studies on CVA6 identified three particles - mature virions, uncoating intermediates (A), and empties. This study reports that the predominant CVA6 particle is the mature virion, despite earlier claims that the A-particle was the primary infectious form and represented the prime target for vaccine development; establishes KRM1 as the uncoating receptor; and that HSPG is the primary attachment receptor. Two potent CVA6-specific neutralizing monoclonals target a newly identified antigenic site located near the three-fold axis of viral capsid that blocks the KRM1-virion interface. The differences observed in the two Mab binding interactions, including steric effects and binding kinetics, are useful insights. The cryoEM studies appear to be clear and convincing, although there is a step from density map to model that lacks evidence of the goodness of fit, such as regions of the model embedded in transparent density to show that sidechains are well accommodated – these could be included in existing supplementary figures. I do note the inclusion of sections through the density maps that is a helpful indicator of density quality. Overall, this is a valuable study that offers clarity in understanding CVA6 host entry as well as a clear framework for therapeutic antibody intervention. With some consideration on the density-model transition, and a few minor points below, I recommend this manuscript for publication in Nat Comms.

Minor Points

* Line 23: “Cryo-EM shows CVA6-HeB primarily exists as mature virions.” CVA6-HeB has not been defined in the Abstract, and indeed is not mentioned again until the Results (line 108) where we learn that it is a strain, along with CVA6-141. While there is more information in the Materials and Methods, these do need some introduction rather than an assumption that readers will know what they are.

* I recommend against supplying too many significant digits. Eg:

Line 215: exhibits a compact capsid (309.80 Å) – how to measure to 1/100 Å? I think that 310 Å is already an accommodation as the capsids are not perfectly round spheres but are lumpy and bumpy, leading to no simple definition of diameter, certainly not to 1-Angstrom precision.

Line 222: (RMSD) values of 0.295 Å for mature virions and 0.475 Å

Seriously, to 1/1000 Å ? The third digit is totally meaningless, the second is pretty doubtful, the significance here is 0.3Å and 0.5Å, which are very good fits.

There are other examples that the authors can restate.

* Line 222: Notably, no A-particles were detected in CVA6-HeB samples (Supplementary Fig. 3), contrasting earlier reports of A-particle-dominated CVA6-Gdula and CVA6-141 preparations 8,9.

What is an explanation for this? Strain specific or methodological?

Version 1:

Reviewer comments:

Reviewer #1

(Remarks to the Author)

All my comments have been reasonably addressed and recommend publication of the manuscript.

Reviewer #2

(Remarks to the Author)

The authors have satisfactorily addressed the points raised in my review.

Dear reviewers,

We would like to thank the reviewers for their insightful comments and suggestions. We have responded to each of the points raised by our reviewers and highlighted the changes in yellow in the revised text.

Response to reviewer #1' s comments:

Reviewer #1 (Remarks to the Author):

Referee report:

Molecular mechanisms of receptor recognition and antibody neutralization of coxsackievirus A6

In the offered manuscript Ke et al describe a two receptor entry mechanism for CVA6. CVA' s are a significant concern for human health and a deeper understanding of how these viruses infect humans could lead to the development of prophylactics. The authors further develop and describe the mode of action of two MAbs of potential use to treat CVA6 infections. Overall, this reviewer found the study to be soundly conducted and does not have any major criticisms to offer.

Response: We thank the reviewer for the positive comments. Below are our responses to the points raised by the reviewer.

A1-1. At most, this reviewer is curious if the authors could comment on whether their antibodies can also neutralize related KRM1 binding CVs? Having a broad-spectrum therapeutic would raise the overall impact of the story.

Q1-1: We thank the reviewer for raising this important point regarding the potential for broad-spectrum neutralization.

To address this directly, we specifically evaluated the cross-neutralizing capacity of MAbs 1F4 and 3H7 against a panel of related KRM1-using enteroviruses. The data, now presented in **Supplementary Figure 15a**, clearly demonstrate that neither MAb 1F4 nor 3H7 is able to neutralize any of the other KRM1-using enteroviruses tested (including CVA2, CVA3, CVA4, CVA5, CVA8, and CVA12), even at the highest concentration of 10 µg/mL. Furthermore, and as initially shown in **Fig. 1a**, neither MAb neutralized CVA10 strain S0148b even at the highest concentration tested (10 µg/mL), confirming their specificity for CVA6. The molecular basis for this high specificity is revealed by our sequence analysis in **Supplementary Figure 15b**. The alignment shows that the key capsid residues critical for binding by MAbs 1F4 and 3H7 are not conserved among other KRM1-using enteroviruses.

While this indicates that 1F4 and 3H7 are not broad-spectrum therapeutics, this

high specificity is a crucial finding. It definitively shows that the epitopes targeted by our antibodies are unique to CVA6, making them highly valuable as precise tools for CVA6-specific diagnosis and pathogenesis studies.

We have addressed this point by adding the following paragraph to the Discussion section (Lines 553-564): “Our data demonstrate that MAbs 1F4 and 3H7 are highly specific for CVA6. Neutralization assays confirmed that neither antibody exhibited cross-neutralizing activity against a panel of related viruses, including CVA10 (Fig. 1a) and other KRM1-using enteroviruses such as CVA2, CVA3, CVA4, CVA5, CVA8, and CVA12, even at the highest concentration of 10 µg/mL (Supplementary Fig. 15a). The molecular basis for this high specificity is revealed by our sequence analysis, which shows that the key capsid residues constituting the 1F4 and 3H7 epitopes are not conserved among other KRM1-using enteroviruses (Supplementary Fig. 15b). While this indicates that 1F4 and 3H7 are not broad-spectrum therapeutics, this high specificity is a crucial finding. It definitively shows that the epitopes targeted by our antibodies are unique to CVA6, making them highly valuable as precise tools for CVA6-specific diagnosis and pathogenesis studies.”.

Supplementary Figure 15. Specificity of CVA6-neutralizing MAbs 1F4 and 3H7. (a) MAbs 1F4

and 3H7 show no neutralization activity against a panel of related KRM1-using enteroviruses at concentrations up to 10 µg/mL, demonstrating they are CVA6-specific. (b) Sequence alignment reveals the molecular basis for specificity: key binding residues for 1F4 and 3H7 (colored) are not conserved across the capsid proteins of CVA6 (strain 141), CVA2 (HN), CVA3 (XZ), CVA4 (FJ), CVA5 (ZZ285), CVA8 (GS14), CVA10 (S0148b), and CVA12 (Texas). Dots represent residues identical to those of CVA6, and red dashes are gaps.

Minor comments:

Q1-2. Line 61: which stabilizes the capsid.

Requires citation.

A1-2: We thank the reviewer for this suggestion. As recommended, we have added citation 8 (Buttner et al., 2022) to the statement on Line 61. The added reference is: Buttner, C. R. et al. Cryo-electron microscopy and image classification reveal the existence and structure of the coxsackievirus A6 virion. *Commun Biol* 5, 898 (2022). The revised text is as follows: "In mature virions, the VP1 hydrophobic pocket beneath the canyon contains a lipid molecule termed pocket factor, which stabilizes the capsid⁸".

Q1-3. Line 694: which were sharpened with DeepEMhancer

Can the authors comment for what purpose the DeepEMhancer maps were used? While it is acceptable to use modified maps to aid model building, the final model should be refined against the unsharpened maps. Moreover, based on the deposition codes, it isn't clear if the authors have deposited just the sharpened maps or if they have also deposited the unsharpened half maps. This reviewer encourages that all maps that were used in the interpretation of the densities be deposited (eg, unsharpened averaged map, globally sharpened, map filtered by local resolution (if it exists and was used), and deepEMhanced maps).

A1-3: We thank the reviewer for these careful and constructive comments. In the original manuscript, we did not clearly state whether the deposited maps were sharpened. To clarify, all density maps deposited in the EMDB are unsharpened, and their processing workflow is detailed in Supplementary Figs. 3, 6, 11 and 12. All structural analyses in this study were performed using these unsharpened maps. The DeepEMhancer-sharpened maps were used solely to aid in model building, while the final model refinement and validation were conducted against the unsharpened maps. Since the DeepEMhancer maps were not used for final structural interpretation, they were not deposited in the EMDB. The unsharpened maps are accessible under the provided accession numbers.

To avoid ambiguity, we have revised the manuscript as follows:

In the Materials and Methods section (Lines 726-729), the text now reads: “Subsequent iterative heterogeneous, homogeneous, or non-uniform refinement was performed, and the resulting density maps were used for model building, refinement, validation, and structural interpretation. The maps were further sharpened using DeepEMhancer to facilitate model refinement”.

In the Data Availability section (Lines 795-796), we now state: “The corresponding unsharpened cryo-EM density maps have been deposited in the Electron Microscopy Data Bank (EMDB).”.

Q1-4. Figure 3(R-T), Line 966 and Figure 5(G-H), Lines 1001-1002: Please could the authors state if the central sections through the maps are masked and sharpened or not.

A1-4: We thank the reviewer for this question. The central section shown in the maps are neither masked nor sharpened, as the application of masks or sharpening filters would alter the native density and potentially lead to a loss of signal in the central regions. We have revised the respective figure legends to explicitly state this point. The legend for Figure 3 (and correspondingly for Figure 5) now includes the following clarification: “Central sections are displayed without masking or sharpening to preserve the native density information.”.

Q1-5. Figure 4A-B: Are the represented densities from sharpened maps or deepEmHanced ones?

A1-5: We thank the reviewer for raising this point. The cryo-EM densities shown in Figure 4a and 4b are from the final, unmodified map (EMD-6503, 2.55 Å) resulting from homogeneous refinement, as outlined in Supplementary Figure 6. This map was not sharpened or processed with DeepEMhancer for the presentation in this figure. In response to your comment, we will update the Figure 4 legend to include the following note for absolute clarity: "Density maps displayed are unsharpened.".

Q1-6. Figure 5 and 6 (E-F): The cartoon representations could do with additional captioning to aid readers. Eg. indicate where the 5-fold axis is, highlighting re-organisation of the loops involved in the binding of the pocket factor.

A1-6: We thank the reviewer for this helpful suggestion. We have revised the cartoon representations in Figures 5 and 6 (E-F) as recommended by:

1. Indicated the 5-fold axis: added a clear annotation in each figure to mark the location of the icosahedral 5-fold axis.

2. Highlighted the reorganized loops: the key VP1 GH loop involved in pocket factor binding is now highlighted to draw attention to its conformational change.
3. Enhanced the figure legends: the legends for both figures have been updated to explicitly describe these additions and the structural consequences.

The updated panels (E-F) for both figures and their corresponding legends are presented below for your review. We believe these additions significantly improve the clarity for readers.

Fig. 5. (e-f) Atomic models of the VP1 hydrophobic pocket in the 1F4-bound virion (**e**) and empty particle (**f**). The pocket factor (magenta) is present in the 1F4-bound virion but absent in empty particle. Consequently, the VP1 pocket collapses in empty particle, accompanied by a significant inward shift of the VP1 GH loop (residues 223–225; red arrowhead, RMSD = 1.4 Å). 1F4 Fab is omitted in (**e**) for clarity. The five-fold axis is indicated.

Fig. 6. (e-f) Structures of VP1 hydrophobic pocket in the 3H7-bound virion (**e**) and 3H7-associated empty particle (**f**). A pocket factor (magenta) is present in the virion but absent in the empty particle, leading to pocket collapse and an inward displacement of the VP1 GH loop (red arrowhead). 3H7 Fab is omitted in both panels for clarity. The five-fold axis is indicated.

Response to reviewer #2' s comments:

Reviewer #2 (Remarks to the Author):

The manuscript “Molecular mechanisms of receptor recognition and antibody neutralization of coxsackievirus A6” by Xianliang Ke et al details the likely host entry mechanisms of CVA6 involving both heparan sulfate proteoglycan (HSPG) for attachment and receptor Kringle-containing transmembrane protein 1 (KRM1) to trigger uncoating. This 2-step process is unlike another coxsackievirus that uses KRM1 for both and is highly relevant for developing antivirals given the recently recognized role of CVA6 as a leading causative agent of hand, foot, and mouth disease. Previous structural studies on CVA6 identified three particles - mature virions, uncoating intermediates (A), and empties. This study reports that the predominant CVA6 particle is the mature virion, despite earlier claims that the A-particle was the primary infectious form and represented the prime target for vaccine development; establishes KRM1 as the uncoating receptor; and that HSPG is the primary attachment receptor. Two potent CVA6-specific neutralizing monoclonals target a newly identified antigenic site located near the three-fold axis of viral capsid that blocks the KRM1-virion interface. The differences observed in the two Mab binding interactions, including steric effects and binding kinetics, are useful insights.

Response: We thank the reviewer for the positive comments. Below are our responses to the points raised by the reviewer.

Q2-1. The cryoEM studies appear to be clear and convincing, although there is a step from density map to model that lacks evidence of the goodness of fit, such as regions of the model embedded in transparent density to show that sidechains are well accommodated – these could be included in existing supplementary figures. I do note the inclusion of sections through the density maps that is a helpful indicator of density quality. Overall, this is a valuable study that offers clarity in understanding CVA6 host entry as well as a clear framework for therapeutic antibody intervention. With some consideration on the density-model transition, and a few minor points below, I recommend this manuscript for publication in Nat Comms.

A2-1: We thank the reviewer for this positive comment and constructive suggestion. In direct response to this comment, we have now added new supplementary figures (Supplementary Figures 4, 7, 13; shown below for your convenience) that showcase the side-chain level model-to-map fit for key regions, such as antibody-antigen interfaces. Furthermore, to explicitly highlight this evidence throughout the manuscript, we have added descriptive text in the main results section that references these figures. The revisions are as follows:

Line 220-222: "In both maps, densities corresponding to the residue backbones and side chains—particularly the bulky ones—were well resolved and readily identifiable (Supplementary Fig. 4)."

Lines 238-240: "The clear and well-fitted densities of both the residue backbones and side chains enabled the determination of the structures of viral particles at atomic or near-atomic resolution (Supplementary Fig. 7)."

Lines 298-300: "The well-fitted backbones and side chains (Supplementary Fig. 13a–c) allowed us to determine the high-resolution structure of the 1F4–virion complex, which preserves characteristic features of mature virions"

Lines 345-347: "The well-resolved backbones and side chains allowed us to determine the structures of both complexes (Supplementary Fig. 13d–i)."

Supplementary Figure 4. Quality of the cryo-EM density maps for CVA6 particles (related to Figure 3a-b). Representative structural motifs (colored sticks) are shown fitted into their corresponding density maps (gray mesh). UV-inactivated CVA6 virion: (a) VP1 (blue), (b) VP2 (green), (c) VP3 (red). CVA6 empty particle: (d) VP1 (blue), (e) VP2 (green), (f) VP3 (red).

Supplementary Figure 7. Quality of the cryo-EM density maps for the CVA6-KRM1 complex and A-particle (related to **Figure 3i-k**). Representative structural motifs (colored sticks) are shown fitted into their corresponding density maps (gray mesh) for the KRM1-bound CVA6 virion (**a-c**, VP1/blue, VP2/green, KRM1/tomato) and the CVA6 A-particle (**d-f**, VP1/blue, VP2/green, VP3/red).

Supplementary Figure 13. Quality of the cryo-EM density maps for CVA6-antibody complexes (related to **Figures 5–6**). Representative structural motifs (colored sticks) are shown fitted into their corresponding density maps (gray mesh). 1F4-CVA6-virion complex: (**a**) VP1 (blue), (**b**) VP2 (green), (**c**) 1F4 VH domain (cyan). 3H7-CVA6-virion complex: (**d**) VP1 (blue), (**e**) VP2 (green), (**f**) 3H7 VH domain (cyan). 3H7-empty particle: (**g**) VP1 (blue), (**h**) VP2 (green), (**i**) 3H7 VL domain (magenta).

Minor Points

Q2-2.* Line 23: “Cryo-EM shows CVA6-HeB primarily exists as mature virions.” CVA6-HeB has not been defined in the Abstract, and indeed is not mentioned again until the Results (line 108) where we learn that it is a strain, along with CVA6-141. While there is more information in the Materials and Methods, these do need some introduction rather than an assumption that readers will know what they are.

A2-2: We thank the reviewer for this helpful suggestion regarding the need to properly

introduce the CVA6 strains. We have revised the manuscript to provide the necessary definition and context upon first mention.

Specifically, we have:

1. Revised the sentence in the abstract (Line 23) to: "Cryo-EM shows that the CVA6 clinical strain HeB primarily exists as mature virions."
2. Added a detailed introduction for the strain in the introduction (Lines 87-89) to read: "We show that purified viral particles of the CVA6 clinical strain HeB (strain 54203/HeB/2012, isolated from an HFMD patient in Hebei Province, China, 2012) predominantly exist as mature virions."

Q2-3: * I recommend against supplying too many significant digits. Eg:

Line 215: exhibits a compact capsid (309.80 Å) – how to measure to 1/100 Å? I think that 310 Å is already an accommodation as the capsids are not perfectly round spheres but are lumpy and bumpy, leading to no simple definition of diameter, certainly not to 1-Angstrom precision.

Line 222: (RMSD) values of 0.295 Å for mature virions and 0.475 Å

Seriously, to 1/1000 Å? The third digit is totally meaningless, the second is pretty doubtful, the significance here is 0.3Å and 0.5Å, which are very good fits.

There are other examples that the authors can restate.

A2-3: We thank the reviewer for this critical and constructive comment regarding the reporting of excessive significant digits, which we agree does not reflect the actual precision of our measurements. We have thoroughly revised the manuscript to correct this issue throughout, in accordance with the reviewer's recommendation.

The sentences cited by the reviewer have been revised as follows (and similar changes have been made globally):

"The CVA6-HeB mature virion structure (2.52 Å resolution) exhibits a compact capsid (~310 Å in diameter) ... The CVA6-HeB empty particle structure (3.47 Å resolution) shows expanded architecture (~317 Å in diameter)."

"Structural alignment with CVA6 prototype strain Gdula⁸ confirmed structural conservation, with overall root-mean-square deviation (RMSD) values of 0.3 Å for mature virions and 0.5 Å for empty particles..."

Q2-4: * Line 222: Notably, no A-particles were detected in CVA6-HeB samples (Supplementary Fig. 3), contrasting earlier reports of A-particle-dominated CVA6-Gdula and CVA6-141 preparations 8,9.

What is an explanation for this? Strain specific or methodological?

A2-4: We thank the reviewer for raising this question regarding the potential causes

for the divergent particle composition observed between CVA6-HeB and the previously reported Gdula and 141 strains.

To address this point directly, we have now added a paragraph in the Discussion section that provides our interpretation of this discrepancy. The added text (lines 506-514) reads: “This discrepancy is likely attributable to a combination of strain-specific characteristics and methodological variations. Intrinsic differences between isolates, such as capsid stability and susceptibility to uncoating, may account for the divergent particle composition. Furthermore, extrinsic factors including cell culture systems, purification protocols, and the criteria for selecting gradient fractions could also contribute to the observed variance. Future studies employing standardized methods to comparatively analyze these strains will be crucial to delineate the underlying causes.”.

We believe this addition provides a balanced and plausible explanation for your question, acknowledging the roles of both biological and technical factors, and points toward the necessary future work to reach a definitive conclusion.